



# Influence of vessel characteristics and atmospheric processes on the gas and particle phase of ship emission plumes: In–situ measurements in the Mediterranean Sea and around the Arabian Peninsula

5  Siddika Celik[1], Frank Drewnick[1], Friederike Fachinger[1], James Brooks[2], Eoghan Darbyshire[2], Hugh Coe[2], Jean–Daniel Paris[3], Philipp G. Eger[4], Jan Schuladen[4], Ivan Tadic[4], Nils Friedrich[4], Dirk Dienhart[4], Bettina Hottmann[4], Horst Fischer[4], John N. Crowley[4], Hartwig Harder[4], and Stephan Borrmann[1,5]

[1]Particle Chemistry Department, Max–Planck–Institute for Chemistry, Mainz, 55128, Germany
10  [2]Centre for Atmospheric Science, University of Manchester, Manchester, M13 9PL, UK
[3]Laboratoire des Sciences du Climat et de l'Environnement, CEA–CNRS–UVSQ, Gif–sur–Yvette, 91191, France
[4]Atmospheric Chemistry Department, Max–Planck–Institute for Chemistry, Mainz, 55128, Germany
[5]Institute for Atmospheric Physics, Johannes Gutenberg University Mainz, Mainz, 55128, Germany

*Correspondence to*: Frank Drewnick (frank.drewnick@mpic.de)

15  **Abstract.** 252 emission plumes of ships operating in the Mediterranean Sea and around the Arabian Peninsula were investigated using a comprehensive dataset of gas and submicron particle phase properties measured during the two–month shipborne AQABA field campaign in summer 2017. The post–measurement identification of the corresponding ship emission events in the measured data included the determination of the plume sources (up to 38 km away) as well as of the plume ages (up to 115 min) and was based on commercially available historical records of the Automatic Identification System. The dispersion lifetime of chemically inert $CO_2$ in the ship emission plumes was determined as $(70 \pm 15)$ min, resulting in levels indistinguishable from the marine background after $(260 \pm 60)$ min. Emission factors (EFs) as quantities that are independent of plume dilution were calculated and used for the investigation of influences on ship emission plumes caused by ship characteristics and the combustion process as well as by atmospheric processes during the early stage of exhaust release and during plume aging. Combustion efficiency and therefore emission factors of black carbon and $NO_x$ were identified to depend mostly on the vessel speed and gross tonnage. Moreover, larger ships, associated with higher engine power were found to use fuel with higher sulfur content and have higher gas phase $SO_2$, particulate sulfate, particulate organics and particulate matter EFs. Despite the independence of EFs on dilution, significant influence of the ambient wind speed on the particle number and mass EFs was observed that can be traced back to enhanced particle coagulation in case of slower dilution and suppressed vapor condensation on particles in case of faster dilution of the emission plume. Atmospheric reactions and processes in ship emission plumes were investigated that include $NO_x$ and $O_3$ chemistry, gas–to–particle conversion of $NO_x$ and $SO_2$ and the neutralization of acids in the particle phase through the uptake of ambient gas phase ammonia, the latter two of which cause the inorganic particulate content to increase and the organic fraction to decrease with





increasing plume age. The results enable identification of the most important influences on (or processes in) ship emission plumes and to describe those quantitatively by parameterizations, which could be used for further refinement of atmospheric models.

## 1 Introduction

Globalization and international trade as one of its essential components are expected to grow in the longer term. Currently, ships are carrying about 90 % of the world's trade by volume (Smith et al., 2015). This is mainly due to cost reasons, because ships are considered as energy efficient means of transport of passengers and goods in comparison to on–road vehicles and aircraft (Jonsson et al., 2011). Thus, shipping represents an indispensable and expanding mode of transport as an international economic factor (Chen et al., 2018).

Historically, there was no extensive, legislative regulation of air pollutant emissions caused by maritime traffic, as (ocean–going) ships operate most of their time far away from areas of human habitation (Juwono et al., 2013; Moldanová et al., 2013). Therefore, especially large ocean–going vessels burn inexpensive residual diesel fuel as cost saving measure (Juwono et al., 2013). This fuel usually has a high sulfur content of up to 4.5 wt. %, whereas the sulfur content of diesel fuel used for the road traffic in most European countries is 4500 times lower, i.e. 10 ppm(m) (Fridell et al., 2008). In this context, since

1973 the International Maritime Organization (IMO) sets binding and periodically tightened regulations for maritime pollutant emissions with the MARPOL convention. Special legally binding restrictions for the emissions of $NO_x$ (= $NO$ + $NO_2$) and $SO_x$ (= $SO_2$ + $SO_3$) as well as of particulate matter (PM) caused by ships exist for so–called Emission Control Areas (ECAs) (IMO, 2019). However, concerning environmental protection these regulations are not yet extensive enough. On a global scale approximately 2.2 % of anthropogenic $CO_2$ emissions, 15 % of anthropogenic $NO_x$ emissions and 5 to 8 %

of anthropogenic $SO_x$ emissions are related to ocean–going vessels (Corbett et al., 2007; Nunes et al., 2017; Metz et al., 2007).

Considering the significant anthropogenic influence of shipping on the atmosphere on a regional and global scale and the associated negative influences on air quality and climate, the investigation of the gas and particle phase of ship emission plumes gains increasing importance from both ecological and epidemiological points of view (Diesch et al., 2013; Juwono et

al., 2013). In the literature, experimental studies can be classified as follows:

- Exhaust sampling of a large number of operating vessels from stationary (e.g. at harbors or shore sites) or mobile (e.g. research vessel or aircraft) measurement platforms (e.g. Williams et al., 2009; Jonsson et al., 2011; Alföldy et al., 2013; Diesch et al., 2013);
- Multiple sampling and tracking of the emissions of few individual operating vessels using a research vessel or

aircraft (e.g. Sinha et al., 2003; Chen et al., 2005; Cappa et al., 2014);
- On–board studies conducted on an operating vessel (e.g. Agrawal et al., 2008; Moldanová et al., 2009, 2013);
- Measurements with a test rig in a laboratory (e.g. Petzold et al., 2008).





These studies give a broad overview of emission factors (EFs) and their variability for a variety of species of the aerosol gas and particle phase. EFs refer the quantity of emitted species to the amount of burned fuel and are therefore independent of the emission plumes' dilution. However, so far these investigations neither provide an extensive knowledge of the parameters causing the observed variability nor on the extent to which these parameters affect the emissions. On–board and

test rig studies could be partially exempted from this statement, but they do not offer information on ship emission plumes on a large selection of ships or under real–world conditions, respectively. Moreover, previous experimental studies provide little information on atmospheric processing and aging of ship emission plumes because in most cases ship emission plumes were investigated that were emitted at less than 5 km distance and were less than 15 min old when reaching the measurement site (Alföldy et al., 2013; Diesch et al., 2013). In addition, the measurement conditions in these studies (e.g. time of day,

meteorology, ships' operating conditions, etc.) only vary to a minor extent. Furthermore, previous studies usually cover a small number of species in ship emission plumes, which makes it difficult to investigate processes running in parallel that can be traced to the same cause (e.g. to atmospheric aging or to the combustion process).

The AQABA (Air Quality and climate change in the Arabian BAsin) field campaign provides an extensive and diverse dataset to investigate ship emission plumes, emitted at various horizontal distances to the measurement location and

therefore of different ages. For a variety of species of the aerosol gas and particle phase data have been collected on–board the research vessel *Kommandor Iona* during this two–month field campaign in summer 2017. The research vessel's route led from Southern France through the Mediterranean Sea and around the Arabian Peninsula to Kuwait and the same way back. Given the large geographic and temporal coverage of the field campaign the dataset contains information on emissions of a diversity of vessels of different size, speed, type, etc. that were measured under various atmospheric conditions regarding

meteorology and solar radiation.

The aim of this study is to investigate ship plumes emitted in the Mediterranean Sea and around the Arabian Peninsula using time–resolved measurements of aerosol gas and particle phase species that were recorded during the AQABA field campaign. For this purpose, ship emission plumes were determined and characterized according to their age, transport distance, and source vessel using historical Automatic Identification System (AIS) data. Plume quantities of identified ship

emission plumes, especially EFs, were calculated in order to investigate them with respect to the following influencing parameters:

- **Fuel quality**: How does the fuel sulfur content influence the quantity of emitted species?
- **Ship parameters and combustion process**: How does e.g. the ship velocity (and thus the engine load) affect the combustion efficiency? What are the effects of the exhaust system's size and thus of the exhaust's residence time in

it?

- **Meteorology and solar radiation**: How do atmospheric dilution, transport, and aging affect the characteristics of ship emission plumes?



## 2 Experimental methods

### 2.1 AQABA field campaign

The AQABA field campaign took place from 24 June to 03 September 2017 on the research vessel *Kommandor Iona* that served as measurement platform. Figure 1 shows the route of the research vessel during the field campaign. The first leg started at the seaport of Toulon, proceeding consecutively through the Mediterranean Sea, the Suez Canal, the Red Sea, the Gulf of Aden, the Arabian Sea, the Gulf of Oman and the Persian Gulf and ended on 31 July when reaching the harbor of Kuwait. The second leg included the reverse route from Kuwait back to Toulon. Time series for a variety of species of the aerosol gas and particle phase were measured with high time resolution (between 1 s and 1 min depending on the measured variable), including both background air and events from individual ship emission plumes. The main objectives of the AQABA field campaign were to gain knowledge on the influences of natural and anthropogenic emissions on air quality and climate and to investigate the interaction of the terrestrial and marine ecological systems.

### 2.2 Instrumentation for shipborne in–situ measurements during AQABA

For the purpose of the AQABA field campaign five measurement containers equipped with measurement instruments were installed on the *Kommandor Iona* and the sampling of outdoor air was realized through appropriate inlet systems. Table 1 gives an overview of the measurement instruments that were used for this study. In general, the processing of data (timestamp: UTC) included calibrations, corrections (e.g. of the time delay due to the transport time of sampled air through the inlet system), the removal of unusable data (e.g. due to malfunctioning instruments) and the removal of data contaminated by sampling the ship's own exhaust. All presented particle and gas phase data were scaled to a pressure of $p = 1013.25$ hPa and a temperature of $T = 20\ °C$.

**Particle phase measurements**. The instruments used to measure particle phase data are described in detail by Drewnick et al. (2012). For the measurement of aerosol particles, a self–regenerating silica gel aerosol dryer with two parallel, switchable columns was integrated between the sampling inlet, located approximately 12 m above sea level, and the corresponding instruments. One of the columns was in use to dry the sampled ambient aerosol (relative humidity (RH) behind the aerosol dryer was typically ≤ 40 %), while the other one was regenerated at the same time by heating and flushing it with dry air. The sampling through the columns was switched between the two once per day. Particle loss due to the whole inlet system, estimated for the relevant size ranges (see Table 1; von der Weiden et al., 2009), was largely negligible. Concentrations of non–refractory submicron particle chemical components, namely particulate sulfate ($SO_4^{2-}$), particluate nitrate ($NO_3^-$), particulate ammonium ($NH_4^+$), particulate chloride ($Cl^-$), and total particulate organics, were determined with a High–Resolution Time–of–Flight Aerosol Mass Spectrometer (HR–ToF–AMS; DeCarlo et al., 2006) using the medium mass resolution mode ("V–mode") with 30 s time resolution. The ionization efficiency of the instrument as well as the relative ionization efficiencies of $SO_4^{2-}$ and $NH_4^+$ were determined via regular calibrations throughout the campaign. The atomic O/C





and H/C ratios were calculated according to Aiken et al. (2008) using the high–resolution mass spectra. Averaged data with 10 s time resolution were used in case of all following particle phase data. An Aethalometer measured black carbon (BC) concentrations in $PM_1$ via light absorption ($\lambda = 880$ nm) through on–line filter sampled aerosol particles. Due to malfunctions, the instrument was exchanged for an identical one once during the field campaign. A default C–factor of 1.57

was applied to account for multiple scattering within the filter (Drinovec et al., 2015). Temperature fluctuations due to the on– and off–switching of the air conditioning system in the measurement container caused periodic fluctuations of the measured BC concentration. These fluctuations were taken into account by always averaging over full fluctuation periods when calculating average concentrations as described in Sect. 2.4. Polycyclic aromatic hydrocarbon (PAH) mass concentrations on $PM_1$ particles were detected with a PAH monitor. Particle number concentrations (PNCs) are given both

by an ultrafine water–based Condensation Particle Counter (CPC) and by the integrals of the size resolved PNCs over the whole size range of a Fast Mobility Particle Sizer (FMPS). PNCs used for this study were calculated by averaging the data from both instruments. In addition, the FMPS provides the particle number size distributions. As the ship emission plumes measured during the AQABA field campaign showed particles exclusively in the (lower) size range of the FMPS, this data was used to calculate $PM_1$ particle mass concentrations, assuming spherical particles and an average particle density of 1.53

g cm$^{-3}$ calculated using the mass concentrations of AMS species and BC. The calculated $PM_1$ particle mass concentrations were corrected for under–measurement in the upper size range of the FMPS ($> 130$ nm; Levin et al., 2015) by scaling them with a factor of 1.85, which was derived from comparison with size distribution data from a concurrently measuring Optical Particle Counter (OPC, Grimm Model 1.109). Main modes of the number size distributions were found to be exclusively in the lower size range ($< 130$ nm; see Sect. 3.2) and were therefore not affected by the under–measurement in the larger size

bins.

**Gas phase measurements**. CO and $CO_2$ concentrations were detected by means of a Cavity Ring–Down Spectrometer (CRDS) calibrated with cylinders bracketing ambient concentrations and traceable to WMO standards. An Iodide Chemical Ionization Quadrupole Mass Spectrometer (Iodide CI–QMS) was used to measure $SO_2$ concentrations (Eger et al., 2019). An ozone monitor was used to detect $O_3$ concentrations based on light absorption at 254 nm, and a formaldehyde monitor to

detect HCHO concentrations based on the Hantzsch reaction (Nash, 1953). NO and $NO_2$ concentrations were measured with a Two–Channel Chemiluminescence Detector (Two–Channel CLD, modified version of the commercial instrument; see Table 1). $NO_x$ mixing ratios were detected by a Two–Channel Thermal Dissociation CRDS which is a modified version of the instrument described by Thieser et al. (2016). Some of the gas phase data, especially those of the nitrogen oxides, show periodic gaps of 1 to 2 min duration due to periodic background measurements or calibrations. These affected in some cases

the detected ship emission events and, if possible, were taken into account by reconstructing the events using the remaining parts of the times series according to its shape by either linear or Gaussian fits.

**Auxiliary data**. A shipborne automatic weather station provided meteorological quantities (ambient temperature $T$, atmospheric pressure $p$, wind speed, wind direction and RH) as well as via GPS the geographic location of the research vessel. A CCD spectral radiometer provided the photolysis rate constants $JO^1D$ (Meusel et al., 2016).



## 2.3 Identification and determination of ship emission plumes, their sources and ages

Since ships are point sources of pollutant emissions and usually the closest sources on the open sea, transient concentration peaks of a few minutes' duration in the AQABA dataset were considered to be potential ship emission events. The AQABA dataset contains a total of 815 of these potential single ship emission events for which the measurement locations (longitude
and latitude) and measurement times were determined referring in each case to the event maximum. Using this data, commercially available historical AIS data were requested from MarineTraffic (MarineTraffic, 2019) in order to try to identify the vessels that caused the emission events (see Fig. 2). For this purpose, data on all vessels were requested which were in a 40 km × 40 km area around the measurement location during a time frame of ± 15 min around the peak observation time. Two AIS records were requested for each of these vessels, the first one complying with the time and
location restrictions and as close as possible to the measurement time and the second "previous" one being recorded at least 20 min beforehand. The second AIS record was included to check the consistency of a vessel's track within the box determined by its velocity and course (i.e. all of the following calculations were perfomed for both AIS records of each ship candidate and the end results were compared). Besides that, a "previous" vessel position might be closer to the sought emission time as the emission of a plume occurs earlier than its detection. The requested AIS records included the following
information: vessel identification number (Maritime Mobile Service Identity, MMSI), date and time, as well as vessel position, speed and course. From the MMSI numbers of identified ships vessel parameters like vessel name, type, gross tonnage (GT, a dimensionless measure for the vessel size) and engine power were obtained from the databases of MarineTraffic (MarineTraffic, 2019), FleetMon (FleetMon, 2019) and Equasis (Equasis, 2019).

For 648 of the 815 potential ship emission plumes a total of about 20 000 AIS records were available according to the
restrictions. The identification of the vessel corresponding to a potential ship emission event by using these AIS vessel data was done with the aid of a self–developed software tool, which performed all routine calculations (see supplement, Sect. S1) to reduce the manual effort and avoid routine calculation errors.

In a first step, average wind data were calculated for each event using the meteorological quantities recorded on–board the *Kommandor Iona*. An averaging interval of 5 min was applied for the analysis. Additionally, averaging intervals of 3 min
and 10 min were used for checking the consistency of results for the below described calculations; in case of inconsistent average wind data and therefore inconsistent results the potential ship emission event was not further analyzed. The average wind speed was given by the arithmetic mean and the average wind direction was calculated according to directional statistics (Eq. (S1); Mardia and Jupp, 2000). As the raw data points of both the wind speed and direction were considered as error–free, the uncertainty of the average wind speed was given by the standard deviation and the uncertainty of the average
wind direction through Gaussian error propagation of Eq. (S1).

In a next step, the position where the track of the probed air mass (assuming to arrive from the average wind direction) intersects the (past or future) track of the vessel candidate (see Fig. 2) was calculated by means of trigonometric navigational equations (Veness, 2019; Tseng and Chang, 2014; Eqs. (S2) to (S12)). For this purpose, the earth was approximated as a





sphere and the air mass and the potential source vessel were assumed to move on a great circle, respectively (see Fig. S1). The right intersection point (out of the two possible) of these two great circles is the one closer to the measurement location. With this potential emission site (the intersection point calculated as above) the distances to the measurement location and to the record position of the considered AIS data point were calculated by means of the Haversine formula (Eq. (S13); Veness,

2019). Using these distances, the wind and vessel speed, the measurement time and the record time of the AIS data point, the times were calculated when the air mass and the vessel candidate were at the intersection point. The time when the air mass was at the intersection point has an uncertainty (calculated via Gaussian error propagation) which is mainly due to the uncertainty of the wind speed. The main uncertainty associated with the time when the vessel was at the intersection point is related to the uncertainty in the wind direction and is given by the time required for the vessel to pass the calculated wind

sector (wind direction ± uncertainty, white area in Fig. 2). In the case that both time intervals (for air mass and vessel at the intersection point) overlap, the candidate vessel was identified as the source of an emission event and the determination of the age and travel distance of the ship's emission plume was possible. Exceptional cases, namely the case of a stationary vessel (e.g. at anchor), the case of a ship course that was (anti–)parallel to the wind direction and the case of wind velocities close to zero, were taken into consideration (see supplement for details). Individual uncertainties for both the ages and the

transport distances of ship emission plumes were estimated from the uncertainties described above and the discrepancy between the results of these calculations for the two AIS records of the identified ship. In general, smaller uncertainties were found for the transport distance (on average 16 %) than for the age (on average 20 %) as the distance was fixed by the vessel track in one direction, whereas the variability of the wind speed affects directly the uncertainty of the determined age of a ship emission plume.

Out of the initially 815 potential ship emission events it was possible to determine the source vessel for 252 ship emission events (see Fig. 1), whereas 156 of these were sampled during daylight hours and 96 during night–time. There are several reasons for unsuccessful ship identifications: the identification of more than one vessel as potential source in areas with high vessel density, the lack of AIS data on vessels (e.g. in areas where only satellite AIS data with little time resolution are available, because vessels had turned off their AIS signal due to piracy, or because of missing MMSI numbers in case of e.g.

small fishing vessels), or other nearby emission sources (e.g. coastal industrial plants, offshore oil rigs). The identified ship emission plumes had an age between 1 min and 115 min and were transported less than 1 km up to 38 km (see Fig. S3); the most frequently encountered plumes had an age of about 20 min and were transported about 4 km. 22 % of the identified ships were oil tankers (world fleet: 11 %), 14 % bulk carriers (world fleet: 12 %), 19 % container vessels (world fleet: 5 %), 8 % general cargo vessels (world fleet: 21 %) and 37 % other types of vessels (world fleet: 51 %) (data for world fleet from

UNCTAD, 2019). Due to regional economic conditions especially oil tankers and container vessels were observed more frequently compared to the world fleet, whereas small vessels falling into the category of other types of vessels were found less frequently probably because of missing AIS identification.



### 2.4 Quantification of plume characteristics

Quantification of characteristics of ship emission plumes was performed using a software tool that was written to perform all calculations, reducing the manual effort and the risk of mistakes.

The average excess (above atmospheric background) concentrations of the particle number and mass as well as of the species

listed in Table 1 were calculated for the 252 ship emission events that were identified. For each ship emission event, two background intervals, one before and one after the event, and one interval including the event itself, were defined (see Fig. S4 for a graphical illustration), with interval limits set as close as possible to the event and with approximately equal interval lengths. This was achieved by comparing the event in each variable's time series to the event in the time series of the PNC, which is the most complete and significant one regarding the identified ship emission events. Accordingly, the intervals have

the same width for each measured quantity of an emission event but differ in adjustments due to minor time shifts. The background was linearly interpolated between the two background intervals and subtracted from the average event concentration to obtain the average excess concentration in the plume. Periodic fluctuations of the measured BC concentration and data gaps during a ship emission event were taken into account as described in Sect. 2.2. In general, calculated average excess concentrations below the detection limit (defined as three times the standard deviation of the

background divided by the square root of the number of measurement points within the ship emission event interval) were excluded from further analysis.

The average excess concentration of a species or the particle number or mass concentration in an expanding ship emission plume changes with time in the first place due to dilution (Petzold et al., 2008; Kim et al., 2009). Emission factors that are independent of the dilution were calculated following Eq. (1) (Diesch et al., 2013):

$$\mathrm{EF}_x = \frac{[x]}{[\mathrm{CO_2}] \cdot \frac{M_C}{M_{\mathrm{CO_2}}}} \cdot w_C,  \tag{1}$$

where $x$ is the excess quantity of a gas or particle phase species, $M_C/M_{\mathrm{CO_2}}$ the carbon mass fraction in $CO_2$ and $w_C = 0.865$ kg C (kg fuel)$^{-1}$ (Diesch et al., 2013) the mass fraction of carbon in marine diesel fuel. $\mathrm{EF}_x$ is given in g or number per kg of burned fuel, $[x]$ is in μg per m$^3$ in case of mass concentrations ($NO_x$ is given as $NO_2$) and in $10^{12}$ # per cm$^3$ in case of number concentrations, and $[\mathrm{CO_2}]$ is in mg per m$^3$. Here, it is assumed that both $x$ and $CO_2$ experience dilution in the same way and

that the concentration of chemically inert $CO_2$ in an expanding ship emission plume is affected exclusively by plume dilution, so that it remains proportional to the amount of burned fuel (Corbett et al., 1999). Furthermore, it is assumed that fuel carbon is completely emitted as $CO_2$ ($CO_2$ balance method), i.e. that the fraction of other carbon species like CO, $CH_4$, volatile organic compounds (VOCs) and particulate carbon in a ship emission plume is negligible compared to $CO_2$. The amount of $x$ is referred in the EF to the amount of burned fuel via the mass fraction of carbon in marine diesel fuel (Diesch

et al., 2013; Sinha et al., 2003). Strictly speaking, emission factors should refer to the time of emission which is possible only in case of conservative plume characteristics that do not change during plume expansion in the atmosphere (Petzold et





al., 2008). In this study, emission factors are used for the investigation of atmospheric processes (except plume dilution) and aging.

Based on the exponential decrease of the average excess concentration of $CO_2$ during the expansion of the ship emission plumes an average plume dispersion time constant of $(70 \pm 15)$ min was determined (see Fig. S5 for $[CO_2](t)$). From the

decrease in excess $CO_2$ concentration during transport it takes on average $(260 \pm 60)$ min to reach the detection limit of the excess $CO_2$ concentration, i.e. the time required for the plume to become indistinguishable from the background. Chen et al. (2005) report that plume dispersion results in a ship's emission event older than 3 h being indistinguishable from the background level, in good agreement with our findings. According to Petzold et al. (2008) it takes less than 24 h until a ship emission plume is completely mixed with the marine boundary layer (MBL).

Average particle number size distributions of ship emission plumes were calculated proceeding in the same way as for calculating average excess concentrations and using the same background and ship emission event intervals as for the PNC. From log–normal fits to the resulting excess particle number size distributions, in each case the count median diameter (CMD), the geometric standard deviation (GSD) and the width of the distribution, referring to one standard deviation, were obtained. The determination of the width followed Eq. (2) (Hinds, 1999):

$$\text{width} = \text{CMD} \cdot \left(\text{GSD} - \text{GSD}^{-1}\right). \tag{2}$$

We define here the expression "potential photochemical processing" of the ship emission plume as the product of the average measured photolysis rate $JO^1D$ during the plume transport and the plume age $(t)$ $(\rightarrow JO^1D \cdot t)$ in Hz·s, which is used as a proxy for potential photochemical processing of plume components in the atmosphere. The relative uncertainty of this quantity is on average 27 % considering the average relative uncertainty of the plume age and the relative measurement

uncertainty of $JO^1D$. Measured OH concentrations were not used as a measure of photochemical processing due to insufficient data coverage and because they only describe the situation at the research vessel's position and not within the plume. A better measure for photochemical processing would be the modelled OH concentrations along the plume transport path. However, this modelling is well beyond the scope of this work.

Using the mass concentrations of particulate organics and the atomic O/C and H/C ratios for the organic aerosol during

plume and background measurements, the O/C ratios for the plume contribution were calculated (Eq. 5) via the average mass concentrations of oxygen and carbon for both the background (B) and for the ship emission event (emission + background, EB) following Eqs. (3) and (4):

$$[\text{O}] = \frac{[\text{organics}] \cdot m_O}{m_O + m_H + m_C} = \frac{[\text{organics}] \cdot \text{O/C} \cdot M_O}{\text{O/C} \cdot M_O + \text{H/C} \cdot M_H + \text{C/C} \cdot M_C}, \tag{3}$$

$$[\text{C}] = \frac{[\text{organics}] \cdot M_C}{\text{O/C} \cdot M_O + \text{H/C} \cdot M_H + M_C}, \tag{4}$$

$$\text{O/C} = \frac{([\text{O}]_{EB} - [\text{O}]_B)/M_O}{([\text{C}]_{EB} - [\text{C}]_B)/M_C}, \tag{5}$$





where $m_x$ is the total mass and $M_x$ the atomic weight of the respective species $x$. For these calculations we assume that particulate organics consist only of oxygen, hydrogen and carbon. To check for consistency, the H/C ratio was calculated analogously and found to always show the expected reverse behaviour to the O/C ratio.

The fuel sulfur content (wt. % fuel S) was calculated according to Diesch et al. (2013) by using the EFs of $SO_2$ and $SO_4^{2-}$ and

assuming that the fraction of other sulfur species like $SO_3$ and gas phase $H_2SO_4$ is negligible in the ship emission plumes.

The modified combustion efficiency was calculated based on the approximation that the fraction of carbon species like $CH_4$, VOCs and particulate carbon in a ship emission plume is negligible compared to $CO_2$, so that the modified combustion efficiency is given by the ratio of the average excess $CO_2$ concentration to the sum of the average excess concentrations of $CO_2$ and CO (Ward and Radke, 1993).

Frequency distributions of all calculated quantities are given in Fig. S6.

## 3 Results

Parallel measurements of multiple variables which are associated with ship emission plumes and observation of such plumes under very different conditions (e.g. plume age, meteorological conditions, source vessel characteristics, etc.) allow the investigation of factors which might influence the characteristics of ship emission plumes. For this purpose, we investigated

the relationship between plume characteristics and various factors (above mentioned measurement conditions) by correlation analysis. As several different factors can influence individual plume characteristics, the correlation plots always show a relatively strong degree of scatter, which led us to bin them, such that the same number of data points (at least 5 and maximal 32) was included in each bin, resulting in not equidistantly distributed bins. The slope, intercept, and correlation coefficient did not significantly change when using binned data instead of raw data. In case raw data are presented their

relative uncertainties combine the estimated quantification (Sect. 2.4) and measurement uncertainties, whereas in case of binned data error bars include in addition one sigma standard deviations of the data distributions in each bin.

In the following sections we first discuss the influences of ship–related properties on the emission characteristics before we present the influences of the early stage of expansion and of chemical processing during transport. Finally, we use averages of emission factors for comparison with data from the literature.

### 3.1 Influences of ship properties and the combustion process on ship emission plumes

**Influence of combustion conditions.** The ship emission plumes of the AQABA dataset enabled us to extract information regarding the influence of combustion conditions. In general, more efficient fuel combustion results in increased $NO_x$ (especially NO) emissions owing to the enhanced oxidation of atmospheric nitrogen that occurs at higher combustion temperatures. It also leads to decreased soot particle (*here*: BC) emissions as a consequence of more efficient oxidation of

fuel carbon (Corbett et al., 1999; Juwono et al., 2013; Pokhrel and Lee, 2015). The fuel combustion efficiency depends primarily on the oxygen–to–fuel mixing ratio in the combustion chamber (Khalid, 2013). This ratio can be influenced by





various factors. We found a strong dependence of the BC emission factor on the ambient oxygen concentration: With increasing ambient pressure–to–temperature ratio (i.e. increasing oxygen concentration according to the ideal gas law) the BC emission factor decreased by more than a factor of 2 over the range of observations, likely due to improved combustion efficiency (see Fig. 3 (a)).

Another important parameter for combustion efficiency is the combustion temperature: With increasing combustion temperature, combustion is more efficient, but more nitrogen oxides are produced. Furthermore, reduced oxidation of NO to $NO_2$ in the oxygen–deficient exhaust gas, which occurs after efficient combustion in the propulsion system, results in an increase of the NO to $NO_2$ ratio with increasing combustion efficiency (Rößler et al., 2017). According to the literature, a higher vessel speed, engine load, or engine power causes a higher peak combustion temperature in the engine system and

thus more efficient fuel combustion (Cappa et al., 2014; Sinha et al., 2003). In agreement with this, we observe a 3–fold increase in the NO to $NO_2$ ratio and an almost 3–fold increase in the $NO_x$ emission factor over the range of observed vessel speeds from 0 to ~10 m s$^{-1}$ (see Fig. 3 (b)). As a consequence of improved combustion, the BC emission factor also decreased to a similar degree, acquiring at a vessel speed of 10 m s$^{-1}$ about one third of the value at 0 m s$^{-1}$. On–board electric power is usually generated by an auxiliary engine, which results in emissions from stationary vessels ($v_\mathrm{ship} \sim 0$ m s$^{-1}$

) (Pokhrel and Lee, 2015; Williams et al., 2009). For very slow vessels, the relative contribution of this auxiliary engine to the overall emissions becomes especially significant.

While a similar dependence of the $NO_x$ emission factor on the engine power was observed (presented in the supplement in Fig. S7) the dependence of combustion efficiency–related EFs on ship size (gross tonnage) is more complex. While for small ships the combustion efficiency seems to increase (increasing $NO_x$ EF, decreasing BC EF) with increasing gross tonnage,

which was also observed by Diesch et al. (2013) and Williams et al. (2009), a plateau is reached at about a gross tonnage of 50 000, and for very large vessels (GT > 150 000) the combustion efficiency seems to deteriorate again without any obvious reason (see Fig. 3 (c)). For vessels in the optimum size range the better combustion efficiency, compared to the very small ships, results in about 50 % lower BC emissions and more than doubled $NO_x$ emissions per kg of burned fuel.

**Influence of fuel quality.** The influences of fuel quality on ship emission plumes were investigated with regard to fuel sulfur

content. Diesel fuel used for shipping is in general a blend of refined fuel and residual oil (Mohd Noor et al., 2018). For economic reasons, large ocean–going vessels often burn cheap heavy fuel oil (HFO) that is highly contaminated with sulfur, ash, asphaltenes, and metals (Corbett et al., 1999; Buffaloe et al., 2014). High quality but expensive marine gas oil (MGO) and marine diesel oil (MDO) are usually only used for the operation of small vessels and vessels operating in ECAs or in coastal areas (Buffaloe et al., 2014; Cappa et al., 2014). Our dataset suggests a general increase of the fuel sulfur content,

calculated from the emission factors of $SO_2$ and $SO_4^{2-}$, with increasing ship size (see Fig. 4 (a)), and therefore more frequent usage of cheap HFO fuel in the larger vessels.

In order to minimize corrosive effects from the high sulfur content in the fuel, ship engine systems are designed to operate at high exhaust flow rates and temperatures, avoiding formation of $SO_3$, condensation of water vapor and formation of sulfuric acid (Pirjola et al., 2014; Moldanová et al., 2009). Furthermore, lubricating oil serves not only as protection against wear but



also as neutralizer for corrosive species like sulfuric acid in the exhaust system. For these reasons, lubricating oil consumption, and as a consequence emission of organic particulate matter, depends on fuel quality (Lack et al., 2009; Juwono et al., 2013; Cappa et al., 2014; Chu–Van et al., 2018). We found almost a doubling of organic aerosol EFs (and of total PM$_1$ EFs) over the range of observed fuel sulfur contents from 0.5 to 2.5 wt. % (see Fig. 4 (b)), showing the significant

indirect influence of fuel sulfur content on ship emission characteristics.

**Coagulation in the exhaust system.** Generally, coagulation of particles, which reduces particle number concentrations without changing particle mass concentrations, occurs more efficiently at higher particle number concentration levels. Longer residence times within the exhaust system, as they occur in larger ships, should therefore result in lower particle number concentrations due to enhanced coagulation effects. In agreement with this, we observe a reduction in particle

number EF with increasing ship size (see Fig. 5), similar to the observations by Diesch et al. (2013).

### 3.2 Influences during the early stage of expansion of ship emission plumes

**Effect of wind speed.** As previously mentioned, coagulation reduces the particle number concentration more efficiently when the latter is high. This is especially the case during transport of exhaust within the exhaust system of the ships. Emission into ambient air causes a sudden dilution of the exhaust, which quickly quenches coagulation. However, the level

of further coagulation depends on the concentration level in the transported plume, which is expected to depend on the ambient wind speed, as it influences the degree of dilution in this phase of emission. In addition, lower concentration levels result in reduced condensation of vapors in the cooled exhaust gas (Hinds, 1999). Both effects can be clearly observed in our ship emission plume data as shown in Fig. 6: With increasing wind speed (stronger dilution) particle mass emission factors are reduced (less condensation) while particle number emission factors strongly increase (less coagulation). Over the range

of observed ambient wind speeds from about 2 m s$^{-1}$ up to ~12 m s$^{-1}$ we observe a reduction of the PM$_1$ emissions per kg of burned fuel by about 40 %, which is similarly reflected in the EFs of condensing species like sulfate; due to the neutralization of sulfuric acid with ambient ammonia (NH$_3$), the ammonium EF shows the same behaviour (see Fig 6 (a)). However, there is no obvious reason why the EF for BC is also reduced at higher wind speed, indicating that, apart from condensation, another unknown effect might also play a role; potentially reduced apparent BC concentrations due to reduced

co–condensed aerosol mass in the Aethalometer might explain parts of this increase. For the particle number emission factor the dilution effect is even more pronounced: Over this range of wind speeds the number of particles emitted per unit of fuel increases more than 3–fold (Fig. 6 (b)).

The effect of reduced coagulation and condensation at higher wind speed is also reflected in the observed particle size distributions: At higher wind speed the particles in the emission plume are smaller than under less windy conditions (see Fig.

6 (c)). Also the width of the particle size distribution is smaller at higher wind speed and more dominated by the combustion mode particles that, according to Petzold et al. (2008), mainly consist of BC and organic material. Note that in this study we only found apparently monomodal size distributions of particles in the ship emission plumes (see Fig. S8).



### 3.3 Atmospheric processes during transport of ship emission plumes

Due to the large range of observed plume ages, the ship emission plumes of the AQABA dataset contain information about the influence of atmospheric aging on plume characteristics. The investigation of atmospheric processing and aging using EFs and average excess plume concentration ratios is possible as these quantities already account for plume dilution effects. We will mainly focus on gas–to–particle conversion and associated chemical reactions in ship emission plumes during atmospheric aging.

**Nitrogen oxide processing.** Atmospheric processing of emitted nitrogen oxides is driven by $O_3$ as well as OH, $HO_2$ and $RO_2$ radicals and affects both the partitioning between NO and $NO_2$ and the conversion of $NO_x$ into nitric acid and finally nitrate aerosol.

NO, the major $NO_x$ component in ship exhaust systems (ca. 95 %) (Song et al., 2003; Williams et al., 2009), depletes ambient $O_3$ and is, as initial NO concentrations in the exhaust are far larger than ambient $O_3$ concentrations, partially converted into $NO_2$ rapidly after emission from the ship stack, causing $NO_2$ fractions in the order of the former ambient $O_3$ in young plumes. Song et al. (2003) modelled the dispersion and chemical evolution of ship emission plumes in the MBL with a Lagrangian plume model, based on defined background photochemical input parameters and estimates for the ship's $NO_x$ emissions, and observed a rapid establishment of the $NO_2$ to $NO_x$ ratio in ship emission plumes after $O_3$ titration during daylight hours. We observe, that the NO to $NO_2$ ratio decreases quickly down to 0.2 (i.e. one third of its initial ratio) during the first half hour of atmospheric transport of ship plumes emitted during daytime. The photochemical equilibrium seems to be established when reaching this ratio and we do not find a subsequent change of it in the range of transport times covered by this study (up to 115 min, Fig. 7 (a)). The partial recovery of the ozone level over time (see Fig. S9) suggests the presence of reactive organic compounds.

Further oxidation of $NO_2$ in the aging plume involves hydroxyl radicals as well as ozone. Directly after exhaust release the OH radical production is strongly suppressed due to the lack of precursors (e.g. ozone) and increases (during daylight hours) in the aging plume with the recovery of $O_3$ and mixing of MBL air into the plume. In addition, during the first stages of plume aging, depending on local $NO_x$ concentration and therefore on the dilution of the ship exhaust, the much more abundant $HO_2$ radical is converted increasingly to OH without consuming $O_3$ (R1) so that the OH radical concentration in the plume is enhanced (Song et al., 2003).

$$\mathbf{NO + HO_2 \longrightarrow NO_2 + HO} \tag{R1}$$

Besides that, the concentration of $N_2O_5$, present essentially only at night–time, increases with increasing $NO_x$ concentration in the ship emission plume (Song et al., 2003).

$$\mathbf{NO_2 + OH \longrightarrow HNO_3} \tag{R2}$$

$$\mathbf{NO_2 + O_3 \longrightarrow NO_3 + O_2} \tag{R3}$$

$$\mathbf{NO_3 + NO_2 + M \rightleftharpoons N_2O_5 + M} \tag{R4}$$

$$\mathbf{N_2O_5 + H_2O\ (het.) \longrightarrow HNO_3} \tag{R5}$$



These two effects cause the lifetime of $NO_x$ in aging ship emission plumes to be shortened compared to processes in the surrounding MBL during the day (R2) and night (R3 to R5) (Chen et al., 2005; Song et al., 2003). As a consequence of these processes, $NO_x$ is increasingly consumed as the plume ages. We found a removal of almost half of the nitrogen oxides from the emission plume within the observed range of short wavelength UV radiation exposures (potential photochemical

processing; see Fig. 7 (c)); at the same time nitric acid and – depending on partitioning between gas and particle phase – nitrate–containing aerosol is produced (see Fig. S10). Regarding the $NO_3^-$ emission factor (not differentiated between day and night–time data), most data points follow a correlation with very low increase of the $EF_{NO_3^-}$, with generally very low $EF_{NO_3^-}$ possibly due to the high ambient temperatures during this study, favouring the partitioning of this species into the gas phase. However, a second branch with larger slope was found that could not be explained by the available data or

information (including ambient temperature, day/night differences, or measurement location).

**Oxidation of SO$_2$.** As described earlier, diesel engine systems primarily emit fuel sulfur as $SO_2$ so that, in principle, particulate sulfate is not formed until the release of the exhaust into the atmosphere (Pirjola et al., 2014; Moldanová et al., 2009). Following the emission into ambient air, $SO_2$ is oxidized by OH radicals and finally partitions into the particle phase mainly as $H_2SO_4$ (see Reactions (R6) to (R8); Lovejoy et al., 1996), which was concluded in previous studies from the

absence of $NH_4^+$ and a significant fraction of sulfate–bound water in the particle phase (Petzold et al., 2008; Pokhrel and Lee, 2015; Schneider et al. 2005).

$$\mathbf{SO_2 + OH + M \rightarrow HOSO_2 + M} \tag{R6}$$

$$\mathbf{HOSO_2 + O_2 \rightarrow HO_2 + SO_3} \tag{R7}$$

$$\mathbf{SO_3 + H_2O \rightarrow\rightarrow H_2SO_4} \tag{R8}$$

We found the conversion of $SO_2$ to $SO_4^{2-}$ to be promoted by high water vapor concentration, i.e. absolute humidity (see Fig. S11 (a)), which is consistent with Reaction (R8). Conversion of $SO_2$ into sulfate aerosol was observed to reduce the $SO_2$ to $SO_4^{2-}$ ratio by a factor of two within the aging times covered in this study (see Fig. 7 (b)). This is accompanied by a strong increase of the sulfate EF and, due to neutralization with ammonia mixed into the plume during atmospheric dilution, by an increase of the ammonium EF. As a consequence, the $PM_1$ EF increases over time (see Fig. 7 (c)). The formation of

secondary inorganic aerosol during plume transport therefore doubles the initial $PM_1$ emission from the ships within the covered range of potential photochemical processing (see next paragraph for the contribution of secondary organic aerosol). No accumulation of $SO_3$ or $H_2SO_4$ in the gas phase is expected to occur in the process as the calculated weight percentage of fuel sulfur derived from the EFs of $SO_2$ and $SO_4^{2-}$ does not show a discernable dependency on the plume age (see Fig. S11 (b)), i.e. there is no indication for any "missing" sulfur.

**Formation of secondary organic aerosol.** Even though VOCs were measured during the AQABA field campaign, they were not included in the investigation of ship emission plumes because the (potential) ship emission events were not reflected in the corresponding VOC time series. Moreover, unlike the mass of particulate inorganic species in the plume aerosol the mass of particulate organics does not increase significantly during photochemical aging (see Fig. S12). The same



is true for the O/C ratio, which also does not show a clear sign of aging of organic material within the observed range of short wavelength UV radiation exposure (see Fig. 8 (a)). However, the particulate organic EF and the O/C ratio show a positive correlation (see Fig. 8 (b)), generally suggesting an increase of organic particulate mass, potentially through oxidation of gas phase organic material.

**Development of aerosol composition and neutralization of inorganic acids.** As discussed in the sections above, the chemical composition of aerosol particles in ship emission plumes depends on various factors. Combustion conditions mainly influence BC emissions, fuel quality (with sulfur content affecting the consumption of lubricating agent) has an effect on organic aerosol components in the plume and atmospheric processes mainly generate additional secondary inorganic aerosol components, which also partially depend on fuel sulfur content. To investigate the changes in chemical composition

of the plume aerosol particles we calculated average particle compositions for three different plume age intervals including approximately the same number of data points: for plumes younger than 16 min, for plume ages between 16 and 40 min, and for plumes older than 40 min (see Fig. 9 (d)). The processes during plume aging are reflected in these average relative compositions: The organic fraction contributes increasingly less and the inorganic fraction increasingly more to the particle phase. We emphasise that the higher average fraction of BC in $PM_1$ from plumes older than 40 min is not an effect of

atmospheric aging but likely due to slight differences in combustion efficiency. Regardless, BC remains constant within the margins of error. As shown in Fig. 7 (c), the photochemical formation of particulate sulfuric acid is accompanied by an increase in particulate ammonium due to (partial) neutralization of sulfuric and nitric acid, associated with the formation of particulate ammonium (bi)sulfate and ammonium nitrate. This is a consequence of ammonia being mixed into the diluting plume from the marine background air. If we take into account that 3 eq. of $NH_3$ neutralize 1 eq. $H_2SO_4$ and 1 eq. $HNO_3$ in

the particle phase, the average degree of neutralization increases from 61 % for plumes younger than 16 min to 69 % for plumes between 16 and 40 min of age up to 95 % for plumes older than 40 min.

**Other trace gases: formation and degradation of HCHO.** Formaldehyde (HCHO) has previously been observed in ship emission plumes and was explained by enhanced photochemical formation from background $CH_4$ (Lowe and Schmidt, 1983) due to higher OH concentrations in the plume, from VOC oxidation (Marbach et al., 2009), or with direct emission from the

ships due to incomplete combustion (Williams et al., 2009). Our observations show that the HCHO emission factors increase along with photochemical aging of the plume (see Fig. 10), in agreement with photochemical formation processes. The regression intercept (i.e. EF for negligible potential photochemical plume processing), however, is not significantly different from zero HCHO emissions and therefore does not suggest direct emissions of HCHO by the ships (see Table 2).

Atmospheric decomposition of HCHO in the troposphere (R9 to R12) occurs within a few hours under daylight conditions

and leads to the formation of CO in the aging plume (Lowe and Schmidt, 1983).

$$HCHO + h\nu \rightarrow CO + H_2 \tag{R9}$$

$$HCHO + h\nu \rightarrow HCO + H \tag{R10}$$

$$HCHO + OH \rightarrow HCO + H_2O \tag{R11}$$


$$\text{HCO} + \text{O}_2 \longrightarrow \text{CO} + \text{HO}_2 \tag{R12}$$

As a consequence of these processes, we observe an increase of the CO emission factor in the photochemically aging plume (see Fig. 10) which results in apparently 2–4 times higher CO emission factors in the aged plume, compared to the actual emissions from the ship stacks due to incomplete combustion of the fuel. The direct CO emissions from the ships can

therefore only be determined from aged plumes by accounting for this CO increase during plume aging (see correlation function in Table 2).

**Development of particle size and number concentration in the aging plume.** As discussed above, the particle mass in the ship plumes increases mainly due to the formation of $(\text{NH}_4)_2\text{SO}_4$ during atmospheric aging and the mass of particulate organics increases slightly through oxidation. In addition, coagulation of particles in the plume should result in fewer but

larger particles. However, we observe neither a growth of particles nor an increase in the width of the size distribution with increasing plume age (see Fig. 9 (a) and (b)). A reason for this might be that for spherical particles the particle diameter scales only with the third root of the particle mass and therefore the resulting small increase of particle size might not be observable within the large variability due to other influencing factors. Another explanation could be that especially soot particles in freshly emitted diesel exhaust usually show an irregular shape with a fractal dimension well below three, which

may collapse when vapor condenses onto the particles forming less fractal and more compact (more spherical) particles. This effect causes an increase of the particle mass without resulting in an equivalent increase of the measured particle size (*here*: mobility diameter; Pagels et al. 2009).

As illustrated in Fig. 9 (c), we do not observe a dependence of the particle number EF on the plume age during daylight hours, whereas a decrease is observed for ship emission plumes that were detected during night–time. In general, there are

mainly two opposite effects that change the particle number concentration in an expanding plume over time (dilution excluded), coagulation and new particle formation. While the particle number concentration decrease during night–time transport could be due to coagulation within the plume. This effect could be outbalanced or at least superimposed by new particle formation during daytime. However, the large scatter in the daytime particle number EF shows that there are strong additional factors influencing this plume property and detailed microphysical modeling studies are needed to disentangle the

different processes and elucidate their relative importance.

### 3.4 Average ship emission plume quantities and comparison with literature data

For comparison with data available in the literature, we calculated average emission factors, particle size distribution data and ratios of plume components for all ship emission plumes of the AQABA dataset (Table 3). Diesch et al. (2013), Lack et al. (2009) and Williams et al. (2009) as well as Jonsson et al. (2011) report the results of studies conducted in ECAs that

include a large sample of measured ship emission plumes (> 100), whereas Petzold et al. (2008) (in ECA), Sinha et al. (2003) (not in ECA), and Juwono et al. (2013) (not in ECA) investigated individual ship emission plumes of a small number of ships (one or two) under various operating conditions.





In general, the average particle mass emission factors (both for total PM and for individual species) from this study are significantly higher (ca. factor 1.5 to 4) than those previously reported in the literature, which is primarily due to the fact that the AQABA field campaign did not take place in an ECA and that a significant number of atmospherically aged and processed ship emission plumes with thus higher particle mass concentrations (see Sect. 3.3) were analyzed. Therefore,

particle mass emission factors that are inferred for the time of emission (fitting parameters $a$ in Table 2) are lower than average EFs and thus closer to (but still somewhat above) listed literature data. The average BC emission factor determined for the identified ship emission plumes of the AQABA dataset is in good agreement with the data given by Lack et al. (2009). Buffaloe et al. (2014, not included in Table 3) focused on the investigation of BC in 135 individual emission plumes of 71 different ships operating off the Californian coast (an ECA) and report a lower average BC emission factor of $(0.31 \pm$

$0.31)$ g (kg fuel)$^{-1}$, though still higher than EF$_{BC}$ derived from other studies (Table 3) which are more in the range of 0.2 g (kg fuel)$^{-1}$. As we have shown (Sect. 3.1, Fig. 3), EF$_{BC}$ strongly depends on the gross tonnage of the emitting vessel, with the lowest EF$_{BC}$ found for a gross tonnage of ~50 000 with ~0.25 g (kg fuel)$^{-1}$, which is closer to the range observed in previous studies. Therefore, the observed differences could be due to different fleet compositions in the various studies, although other influences like different average ship speeds (compare Fig. 3 (b)) are also likely to play a role. Unlike the

average particle mass emission factors, the average particle number emission factor from our study is consistent with the listed literature data. As particle coagulation is mainly relevant during the early stage of plume expansion, when the local PNC is high, plume aging on the considered time scale is not a strongly influencing factor for the particle number (PN) emission factor.

As EFs of many trace–gases mainly depend on vessel characteristics and fuel quality, data available in the literature can vary

significantly. Sinha et al. (2003) conducted a study off the Namibian coast and for a tanker that was assumed to burn MDO (0.1 wt. % fuel S) they report an SO$_2$ emission factor of $(2.9 \pm 0.2)$ g (kg fuel)$^{-1}$ and for a container vessel that was assumed to burn HFO (2.4 wt. % fuel S) an SO$_2$ emission factor of $(52.2 \pm 3.7)$ g (kg fuel)$^{-1}$. The average SO$_2$ emission factor $((26 \pm 6)$ g (kg fuel)$^{-1})$ and the average weight percentage of fuel sulfur (1.4 wt. % fuel S) of our study are in good agreement with the averaged data of both vessels from their measurements as our study covers a broad range of fuel sulfur

contents. The same statement as for the particle mass emission factors regarding higher values due to plume aging is true for the average HCHO and CO emission factors as the analysis shows that these species were also formed during the aging of the ship emission plumes. The average modified combustion efficiency presented in Table 3 was determined by using the CO emission factors that were referred to the time of emission via the slope of the linear correlation (Table 2) and calculating the average of these values, while for all other values in this table the actually observed concentrations were

used.



## 4 Summary and conclusions

A variety of gas and particle phase species in 252 ship emission plumes measured in the Mediterranean Sea and around the Arabian Peninsula during the two–month shipborne AQABA field campaign in summer 2017 were investigated. For this purpose, a method based on commercially available historical AIS records that include ship position, course and velocity was developed and applied to identify the ship emission plumes and characterize them with respect to their source and age by reconstructing the intersection point of the source vessel track and the track of measured air masses. Observed ship emission plumes were up to 115 min old and transported up to 38 km. Based on the time dependent exponential decay of the average excess plume concentration of the chemically inert $CO_2$ in ship emission plumes a dispersion lifetime of $(70 \pm 15)$ min and a dilution down to a level indistinguishable from background within $(260 \pm 60)$ min were determined. Emission factors (EFs) of the particle number and of particle and gas phase species masses in ship emission plumes were calculated as quantities that account for plume dilution and refer to the amount of burned fuel. These EFs were used to investigate the influences on the emission plume characteristics caused by ship parameters and the combustion process as well as by atmospheric effects during the early stage of exhaust release and during plume aging in order to address the following scientific questions. Figure 11 gives an overview of the observed processes and how they affect plume characteristics.

*How does e.g. the ship velocity affect the combustion efficiency?* We observed more efficient fuel combustion at higher ambient pressure–to–temperature ratios (i.e. higher $O_2$–to–fuel mixing ratios in the combustion chamber) and for ships with higher velocities as well as for vessels with medium size gross tonnages. This was reflected in the BC emission factor being lower and was accompanied by larger $NO_x$ emission factors. Additionally, it was found that more efficient fuel combustion causes a lower initial ratio of NO to $NO_2$. (see Fig. 3)

*How does the fuel sulfur content influence the quantity of emitted species?* We found that larger vessels, that typically burn lower grade fuel, emit more $SO_2$ and, due to the higher consumption of lubricating oil as a corrosion preventive, more organic aerosol and thus promote the formation of more PM during atmospheric processing of the corresponding emission plumes. (see Fig. 4)

*What are the effects of the exhaust system's size and thus of the exhaust's residence time within the pipes?* It was shown that larger vessels emit less particles by number per kg of burned fuel, likely due to their longer exhaust system that favors particle coagulation before the coagulation process is quenched when the exhaust is released into ambient air. (see Fig. 5)

*How do atmospheric dilution, transport, and aging affect the characteristics of ship emission plumes?* We observed that ambient wind speed, which affects the dilution of the emission plume at the ship stack exit, significantly influences the particle number concentration (PNC): A lower wind speed results in less diluted plumes and thus promotes particle coagulation as well as vapor condensation and as a consequence causes a lower PNC EF accompanied by larger particles and a higher particle mass EF. Furthermore, due to the long plume aging intervals covered in this study, we were able to observe and quantify atmospheric reactions and processes like $O_3$ reduction and recovery accompanied by the conversion between NO and $NO_2$, the photochemical formation of HCHO and CO, the formation of secondary aerosol, namely the oxidation of

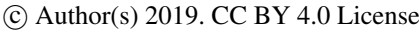

$NO_x$ to $NO_3^-$ and of $SO_2$ to $SO_4^{2-}$, and the neutralization of acids in the particle phase through the uptake of $NH_3$ from ambient air. It was shown that the organic fraction which we mainly trace back to lubricating oil from the ship engine system contributes increasingly less and the inorganic fraction increasingly more to the particle phase with plume aging, although higher O/C ratios were found for higher particulate organic EFs indicating the increase of particulate organics mass through

atmospheric oxidation. (see Figs. 6 to 9)

From this study, the wind velocity was identified as the strongest factor influencing the PN emission factor as well as the median diameter and width of the particle number size distribution. Particulate organics and $SO_2$ emission factors were mainly influenced by fuel sulfur content, strongly associated with vessel gross tonnage. The largest influence on the EFs of

inorganic particle phase species (except BC), the $PM_1$, as well as the CO and HCHO emission factors was found to be atmospheric aging. Finally, the ship velocity was identified to dominate the BC and $NO_x$ emission factors.

The quantification of observed influences on or processes in ship emission plumes by parameterization will help to refine models in this research field. However, detailed numerical simulation studies are required to describe the relevant microphysical and chemical processes quantitatively.

**Data availability**

The data used in this study are archived and distributed through the KEEPER service of the Max Planck Digital Library (https://keeper.mpdl.mpg.de) and will be available from August 2019 to all scientists agreeing to the AQABA protocol.

**Author contributions**

SC performed the analysis of ship emission plumes during the AQABA field campaign and wrote the manuscript. FD

supervised this study. Aerosol particle measurements and data were provided by FD, FF, JB, ED, SB, and HC. JP contributed CO and $CO_2$ measurments and data. $O_3$, $SO_2$, and $NO_x$ measurments and data were provided by PE, NF, and JC and $JO^1D$ values were measured by JS. IT, DD, BH, and HF provided NO, $NO_2$, and HCHO measurements and data. HH took responsibility for the scientific coordination of the field campaign on–board the research vessel. All authors contributed to data interpretation and manuscript revision and approved the submitted version.

**Competing interests**

The authors declare that they have no conflict of interest.



## Acknowledgements

The authors are grateful to T. Böttger, P. Schumann, C. Koeppel, and M. Dorf for technical support in the preparation and conduction of the AQABA field campaign. They thank U. Parchatka for supporting the NO and NO$_2$ measurements during the field campaign. M. Anfield from MarineTraffic is acknowledged for fruitful support in the identification of AIS data. The Max Planck Society covered the majority of the costs for the field campaign; University of Manchester is acknowledged for financial support for data analysis.

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



**Table 1: Overview of quantities measured during the AQABA field campaign that were used in this study, along with the respective measurement instruments, size ranges with the corresponding particle losses, and time resolutions. The given particle losses represent upper limits, as particle losses within the size ranges are lower than at its boundaries. Acronyms are introduced in Sect. 2.2.**

| | variable | measurement instrument | size range (particle losses) | time resolution |
|---|---|---|---|---|
| **particle phase** | PNC | CPC, TSI Model 3787 & FMPS, TSI Model 3091 | 5 nm (20 %) to 1 μm (10 %) & 6 nm (11 %) to 560 nm (0.2 %) [a] | 10 s |
| | PN size distribution | FMPS, TSI Model 3091 | 6 nm (11 %) to 560 nm (0.2 %) [a] | 10 s |
| | $[PM_1]$ [b] | – " – | – " – | 10 s |
| | $[BC]$ | Aethalometer, Magee Scientific Model AE33 | 10 nm (11 %) to 1 μm (3 %) | 10 s |
| | $[PAH]$ | PAH monitor, EcoChem Analytics Model PAS 2000 | 10 nm (11 %) to 1 μm (3 %) | 10 s |
| | $[SO_4^{2-}]$, $[NO_3^-]$, $[NH_4^+]$ $[Cl^-]$, [organics], O/C, H/C | HR–ToF–AMS, Aerodyne Research, Inc. | 40 nm (9 %) to 600 nm (1 %) [c] | 30 s |
| **gas phase** | $[CO]$, $[CO_2]$ | CRDS, Picarro, Inc. Model G2401 | N/A | 1 s |
| | $[SO_2]$ | Iodide CI–QMS | N/A | 10 s |
| | $[O_3]$ | Ozone monitor, 2B Technologies Model 202 | N/A | 10 s |
| | $[NO]$, $[NO_2]$ | Two–Channel CLD, ECO Physics Model CLD 790 SR [d] | N/A | 5 s, 1 min |
| | $[NO_x]$ | Two–Channel TD–CRDS | N/A | 5 s |
| | $[HCHO]$ | Formaldehyde monitor, Aero–Laser Model 4021 | N/A | 3 s [e] |
| **auxiliary data** | longitude, latitude | automatic weather station with GPS, Neptune, Sterela Meteo | N/A | 1 s |
| | wind speed and direction, $T$, $p$, RH | | N/A | 1 s |
| | $J$O$^1$D [f] | CCD spectral radio–meter, Metcon Model 85237 | N/A | 1 min |

[a] Mobility and [c] vacuum aerodynamic diameter.

[b] PM$_1$ covers only particles with diameters ≲ 350 nm, as larger particles were not observed in the ship emission plumes.

[d] Modified version of the commercially available instrument.

[e] Effective time resolution is 170 s.

[f] The photolysis rate was calculated using wavelength resolved actinic flux data measured by the spectral radiometer.





**Table 2: Correlations and corresponding fit parameters and correlation coefficients (Pearson's $R$) referring to dependencies in ship emission plumes presented in Figs. 3 to 10. $N$: number of available raw (not binned) data points.**

| | correlation | | | fit parameters and correlation coefficient | | | |
|---|---|---|---|---|---|---|---|
| | $f(x) = a + bx$ | unit for $f(x)$ | unit for $x$ | $a$ | $b$ | $R$ | $N$ |
| **ship parameters and combustion process** | $EF_{BC}(p \cdot T^{-1})$ | g (kg fuel)$^{-1}$ | hPa·K$^{-1}$ | 29 | −8.5 | 0.37 | 62 |
| | $EF_{BC}(v_{ship})$ | g (kg fuel)$^{-1}$ | m s$^{-1}$ | 1.5 | −0.11 | 0.41 | 63 |
| | $EF_{NO_x}(v_{ship})$ | g (kg fuel)$^{-1}$ | m s$^{-1}$ | 26 | 3.9 | 0.33 | 140 |
| | $[NO]/[NO_2](v_{ship})$ | − | m s$^{-1}$ | 0.076 | 0.032 | 0.18 | 157 |
| | wt. % fuel S(GT) | % | − | 1.3 | 2.1·10$^{-6}$ | 0.16 | 112 |
| | $EF_{organics}$(wt. % fuel S) | g (kg fuel)$^{-1}$ | % | 2.1 | 0.95 | 0.21 | 111 |
| | $EF_{PM_1}$(wt. % fuel S) | g (kg fuel)$^{-1}$ | % | 4.4 | 2.5 | 0.25 | 99 |
| | $EF_{PN}(GT)$ | # (kg fuel)$^{-1}$ | − | 1.3·10$^{16}$ | −2.3·10$^{10}$ | 0.12 | 247 |
| **atmospheric processes early stage** | $EF_{PN}(v_{wind})$ | # (kg fuel)$^{-1}$ | m s$^{-1}$ | 4.5·10$^{15}$ | 1.4·10$^{15}$ | 0.38 | 252 |
| | $EF_{PM_1}(v_{wind})$ | g (kg fuel)$^{-1}$ | m s$^{-1}$ | 9.8 | −0.35 | 0.15 | 156 |
| | $EF_{SO_4^{2-}}(v_{wind})$ | g (kg fuel)$^{-1}$ | m s$^{-1}$ | 4.9 | −0.25 | 0.23 | 165 |
| | $EF_{NH_4^+}(v_{wind})$ | g (kg fuel)$^{-1}$ | m s$^{-1}$ | 1.8 | −0.10 | 0.20 | 76 |
| | $EF_{BC}(v_{wind})$ | g (kg fuel)$^{-1}$ | m s$^{-1}$ | 1.4 | −0.092 | 0.30 | 63 |
| | $CMD(v_{wind})$ | nm | m s$^{-1}$ | 59 | −1.3 | 0.29 | 207 |
| | width($v_{wind}$) | nm | m s$^{-1}$ | 53 | −1.4 | 0.31 | 207 |
| **atmospheric processes aging** | $[NO]/[NO_2](t)$ | − | min | 0.38 | −0.0040 | 0.19 | 157 |
| | $[SO_2]/[SO_4^{2-}](t)$ | − | min | 12 | −0.066 | 0.20 | 125 |
| | $EF_{NO_x}(JO^1D \cdot t)$ | g (kg fuel)$^{-1}$ | Hz·s | 53 | −116 | 0.13 | 139 |
| | $EF_{PM_1}(JO^1D \cdot t)$ | g (kg fuel)$^{-1}$ | Hz·s | 6.5 | 61 | 0.35 | 138 |
| | $EF_{SO_4^{2-}}(JO^1D \cdot t)$ | g (kg fuel)$^{-1}$ | Hz·s | 2.8 | 42 | 0.45 | 149 |
| | $EF_{NH_4^+}(JO^1D \cdot t)$ | g (kg fuel)$^{-1}$ | Hz·s | 0.67 | 20 | 0.60 | 76 |
| | $EF_{NO_3^-}(t)$ (main tendency) | g (kg fuel)$^{-1}$ | min | 0.17 | 0.0070 | 0.58 | 31 |
| | $EF_{organics}$(O/C) | g (kg fuel)$^{-1}$ | − | 2.3 | 4.3 | 0.24 | 152 |
| | $EF_{PN}(t)$ (night) | # (kg fuel)$^{-1}$ | min | 1.5·10$^{16}$ | −1.9·10$^{14}$ | 0.36 | 83 |
| | $EF_{HCHO}(JO^1D \cdot t)$ | g (kg fuel)$^{-1}$ | Hz·s | 0 | 133 | 0.89 | 24 |
| | $EF_{CO}(JO^1D \cdot t)$ | g (kg fuel)$^{-1}$ | Hz·s | 12 | 189 | 0.23 | 111 |





Atmospheric Chemistry and Physics Discussions — Open Access EGU

**Table 3: Overview of average quantities of identified ship emission plumes of the AQABA dataset and comparison with plume quantities presented in the literature. The second row contains a brief study description as measurement site / measurement platform / number of investigated ship emission plumes.**

| | quantity | AQABA 2017 | Diesch et al. (2013) | TexAQS 2006 [e] | Jonsson et al. (2011) | Petzold et al. (2008) [h] | Sinha et al. (2003) | Juwono et al. (2013) |
|---|---|---|---|---|---|---|---|---|
| | | Mediterranean Sea, around Arabian Peninsula / ship / 252 | Lower Elbe river bank / mobile laboratory / 139 | off the Texan coast / ship / > 200 | at Scandinavian harbor / at harbor / 734 | English Channel / aircraft / several of same vessel | off the Namibian coast / aircraft / several of 2 vessels | at Australian harbor / on-board / several of 2 vessels |
| **particle phase** — EF / g (kg fuel)$^{-1}$ | PM | $8 \pm 2$ (PM$_1$) | $2.4 \pm 1.8$ (PM$_1$) [b] | $3.32 \pm 4.04$ (PM$_1$) [b] | $2.05 \pm 0.11$ (PM$_1$) | N/A | N/A | 0.7 to 6.1 (PM$_{2.5}$) |
| | organics | $3 \pm 1$ | $1.8 \pm 1.7$ | $1.26 \pm 0.96$ | N/A | N/A | N/A | N/A |
| | SO$_4^{2-}$ | $4 \pm 1$ | $0.54 \pm 0.46$ | $1.21 \pm 1.50$ | N/A | N/A | N/A | N/A |
| | NO$_3^-$ | $0.8 \pm 0.4$ | N/A | $0.0 \pm 0.1$ | N/A | N/A | N/A | N/A |
| | NH$_4^+$ | $1.3 \pm 0.5$ | N/A | $0.0 \pm 0.1$ | N/A | N/A | N/A | N/A |
| | BC | $0.9 \pm 0.3$ | $0.15 \pm 0.17$ | $0.85 \pm 0.76$ | N/A | $0.174 \pm 0.043$ | $0.18 \pm 0.02$ | N/A |
| | PAH | $0.011 \pm 0.005$ | $0.0053 \pm 0.0047$ | N/A | N/A | N/A | N/A | N/A |
| | EF$_{PN}$ / # (kg fuel)$^{-1}$ | $1.2 \cdot 10^{16} \pm 3 \cdot 10^{15}$ | $2.55 \cdot 10^{16} \pm 1.91 \cdot 10^{16}$ | $1.27 \cdot 10^{16} \pm 0.95 \cdot 10^{16}$ | $2.55 \cdot 10^{16} \pm 1.1 \cdot 10^{15}$ | $1.36 \cdot 10^{16} \pm 2.4 \cdot 10^{15}$ | $4.0 \cdot 10^{16}$ to $6.2 \cdot 10^{16}$ | $1.0 \cdot 10^{16} \pm 2 \cdot 10^{15}$ [f] |
| | CMD / nm | $50 \pm 10$ | N/A [c] | N/A | $29 \pm 3$ [f,g] | N/A [i] | N/A | $90 \pm 10$ [f] |
| | GSD | $1.5 \pm 0.4$ | N/A | N/A | $1.69 \pm 0.04$ [f] | N/A [i] | N/A | $1.62 \pm 0.05$ [f] |
| | width / nm | $50 \pm 10$ | N/A | N/A | N/A | N/A [i] | N/A | N/A |
| | O/C | $0.3 \pm 0.1$ | N/A | N/A | N/A | N/A | N/A | N/A |
| | H/C | $1.6 \pm 0.8$ | N/A | N/A | N/A | N/A | N/A | N/A |
| **gas phase** — EF / g (kg fuel)$^{-1}$ | SO$_2$ | $26 \pm 6$ | $7.7 \pm 6.7$ | $14 \pm 12$ [f] | N/A | 40 to 46 | 2.9 to 52.2 | $1.2 \pm 0.2$ [f] |
| | O$_3$ | $-48 \pm 5$ [a] | N/A | N/A | N/A | N/A | N/A | N/A |
| | NO | $7 \pm 1$ | $16 \pm 12$ | N/A | N/A | N/A | N/A | N/A |
| | NO$_2$ | $35 \pm 6$ | $37 \pm 20$ | N/A | N/A | N/A | N/A | N/A |
| | NO$_x$ (as NO$_2$) | $51 \pm 9$ | $62 \pm 27$ [d] | $68 \pm 25$ [f] | N/A | 96 to 109 | 34.2 to 100.4 [f] | 3.4 to 72 |
| | HCHO | $1.7 \pm 0.8$ | N/A | 0.10 to 0.72 | N/A | N/A | N/A | N/A |
| | CO | $20 \pm 3$ | N/A | $11 \pm 13$ [f] | N/A | N/A | 3.0 to 19.5 | N/A |
| | [CO$_2$] / mg m$^{-3}$ | $1.0 \pm 0.1$ | N/A | N/A | N/A | N/A | N/A | N/A |
| | modified combustion efficiency | $0.995 \pm 0.004$ | N/A | N/A | N/A | N/A | N/A | N/A |
| | wt. % fuel S | $1.4 \pm 0.6$ | $0.38 \pm 0.34$ | N/A | N/A | 2.45 | 0.1 to 2.4 | N/A |

[a] Average reduction of O$_3$ background level during a ship emission event when considering plume dilution and referring to the amount of burned fuel.
[b] PM EF is given by the sum of EFs of AMS species and of BC.
[c] Count mode is on average (35 ± 15) nm; [g] Value is derived from the average geometric mean diameter and the GSD according to Hinds (1999).
[d] NO$_x$ EF was given as sum of EFs of NO and NO$_2$ or [f] as NO EF and converted into NO$_x$ EF as NO$_2$ or [f] by weighting the NO EF with $M_{NO_2}/M_{NO}$.
[e] TexAQS: Texas Air Quality Study. Particle phase data are provided by Lack et al. (2009) and gas phase data by Williams et al. (2009).
[f] Average of available data.
[h] Tracking of one single ship emission plume.
[i] Bimodal size distribution observed.





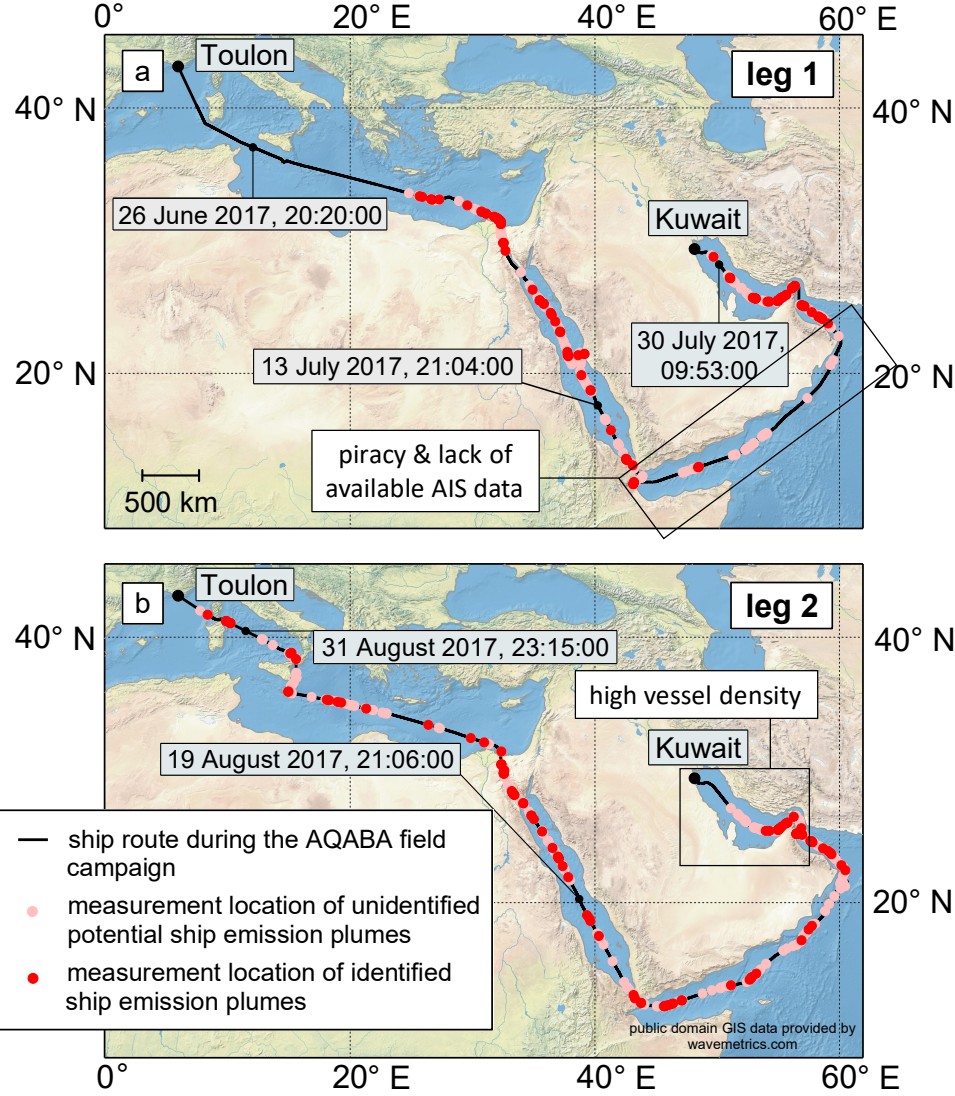

**Figure 1: Route of the research vessel *Kommandor Iona* during the AQABA field campaign on leg 1 (a) and 2 (b) with time labels (UTC) for selected geographic positions. The measurement locations of identified and potential, but unidentified ship emission plumes are marked. The Gulf of Aden and the Arabian Sea as well as the Persian Gulf are indicated as areas where the vessel identification was difficult.**

**Figure 2: Illustration of the method for the identification of ship emission plumes based on an example event. Vessels to be taken into account had to be in a 40 km × 40 km area around the measurement location during a time interval of 30 min considering the plume measurement time as the interval center. Two AIS records were requested after the AQABA field campaign from the AIS data base for each of these vessels, the first one complying with the specified area and time restrictions and the second "previous" one recorded at least 20 min before the first one. Presented are the results of the calculations based on the measurement time and location, the average wind data, their uncertainties and the available AIS records. The areas down– (dark gray) and upwind (light gray) of the measurement location refer to the average wind direction (dashed line; uncertainty is indicated by the white area). The uncertainties of the intersection times and of the corresponding vessel positions are based on the uncertainties of the wind speed. In this specific case, the vessel *Cape Flamingo* (red) was identified as emission source.**

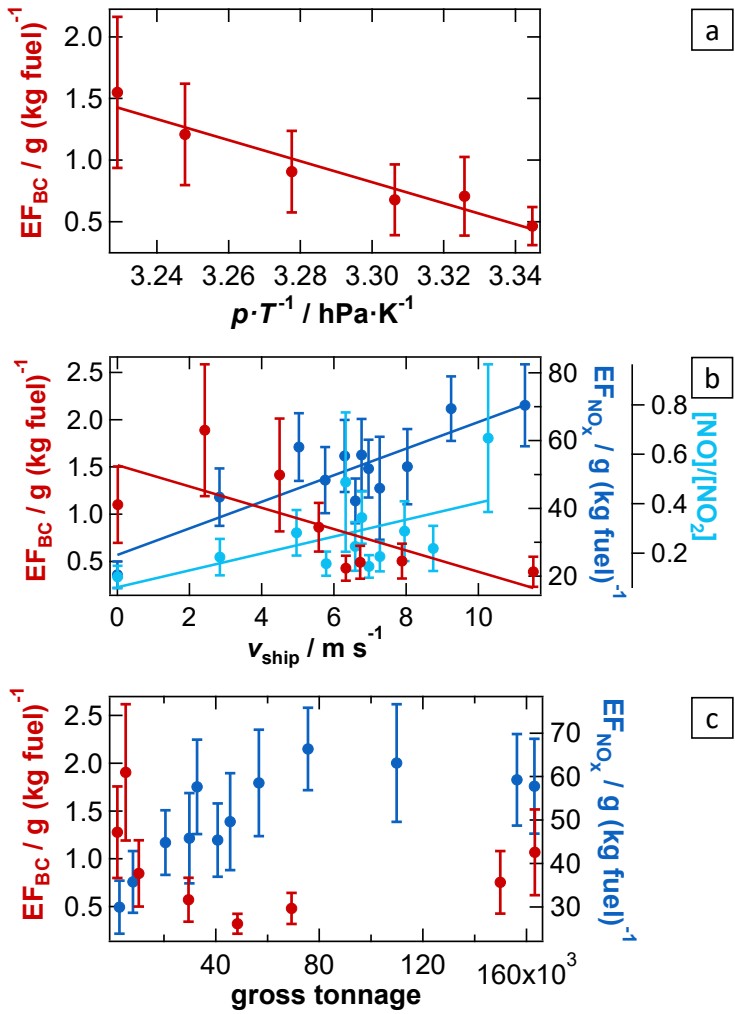

**Figure 3: Influences of combustion conditions on ship emission plumes. The dependency of the BC emission factor on the ratio between atmospheric pressure and ambient temperature is demonstrated (a). Additionally, the dependencies of the NO$_x$ and BC emission factors as well as of the NO to NO$_2$ ratio of average excess concentrations on the vessel speed are presented (b). The correlations between the NO$_x$ and BC emission factors and the vessel gross tonnage are shown (c). Error bars present the combination of estimated quantification and measurement uncertainties and one sigma standard deviations of the data distributions in each bin. The corresponding fit parameters and correlation coefficients are listed in Table 2.**





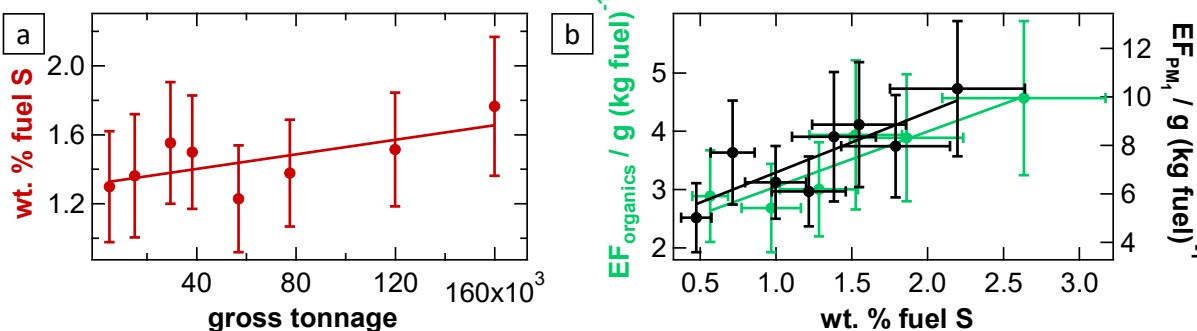

**Figure 4: Influences of fuel quality on ship emission plumes. The correlation between the weight percentage of fuel sulfur and the ships' gross tonnage (a) and the dependencies of the particulate organic as well as the PM$_1$ emission factors on the weight percentage of fuel sulfur (b) are shown. The fit parameters and correlation coefficients are listed in Table 2.**

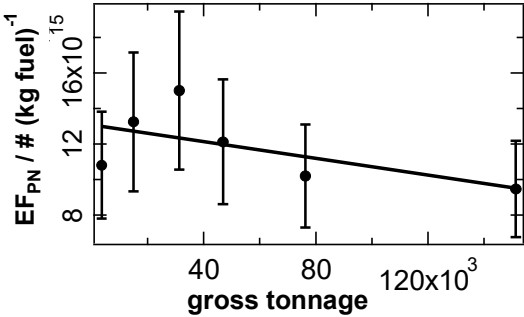

**Figure 5: Influence of the exhaust system's size (in terms of gross tonnage) on ship emission plumes. The particle number EF is plotted versus the gross tonnage of the ship. The fit parameters and correlation coefficient are listed in Table 2.**





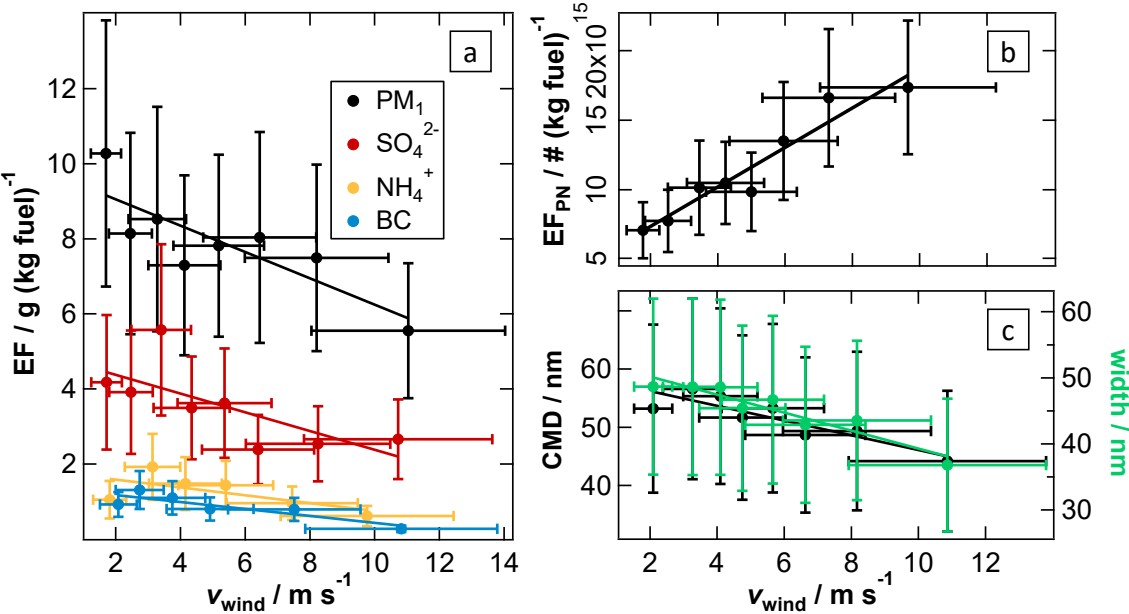

**Figure 6: Effect of dilution during the early stage of plume release on ship emission plumes. PM₁, SO₄²⁻, NH₄⁺, BC (a) and particle number emission factors (b) as well as the CMD and width of the particle number size distribution (c) are plotted versus the wind speed. The fit parameters and correlation coefficients are listed in Table 2.**





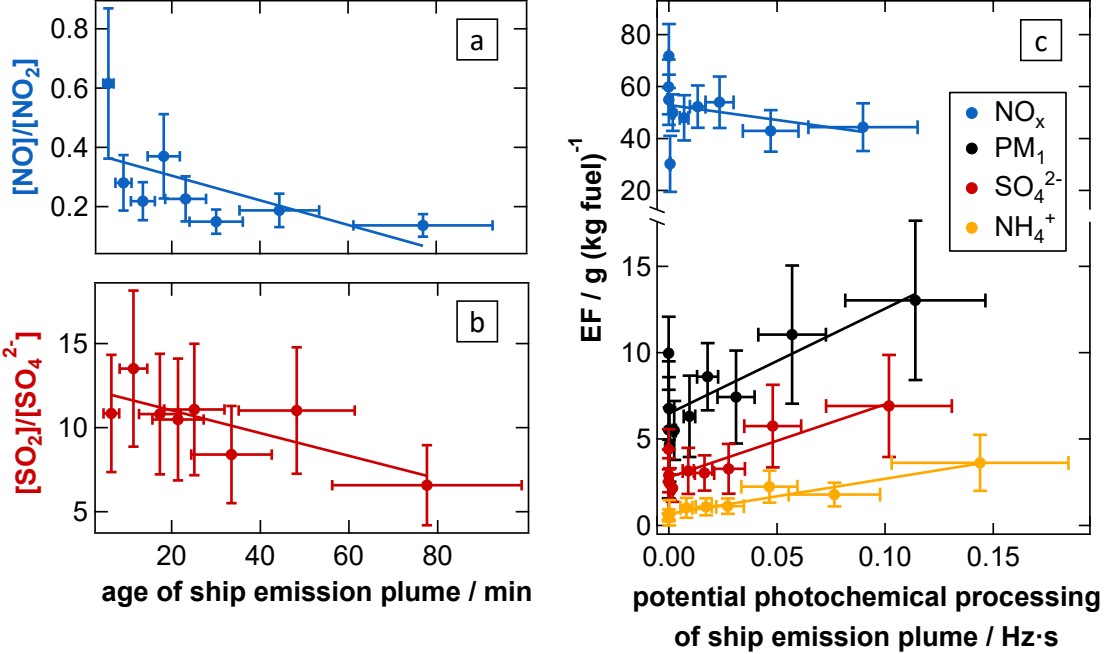

**Figure 7: Atmospheric processing of nitrogen and sulfur species in ship emission plumes. The conversion of NO into NO₂ (a) as well as the conversion of SO₂ into SO₄²⁻ (b) during plume aging are demonstrated through the development of the average excess concentration ratios. Additionally, the decrease of NO$_x$ and increase of PM₁, SO₄²⁻ and NH₄⁺ with increasing potential photochemical processing of ship emission plumes (measured/calculated $J$O¹D integrated over the plume transport time) are shown using the development of the emission factors as indicator (c). The fit parameters and correlation coefficients are listed in Table 2.**

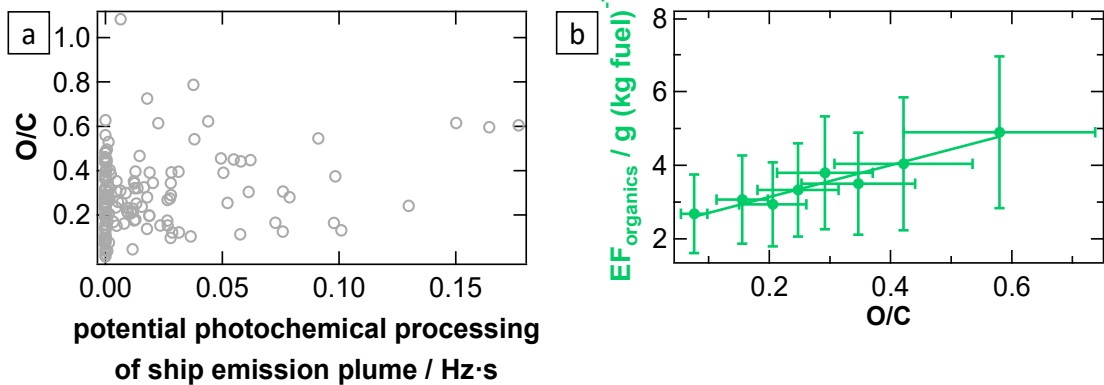

**Figure 8: Development of particulate organics and their oxidation level during atmospheric aging of ship emission plumes. The correlation between the O/C ratio (rel. uncertainty (combined rel. quantification and measurement uncertainties): 48 %) and the potential photochemical plume processing (avg. rel. uncertainty: 27 %; measured/calculated $J$O¹D integrated over the plume transport time) (a) as well as the correlation between the emission factor of particulate organics and the O/C ratio (b) are shown. The fit parameters and correlation coefficient are listed in Table 2.**

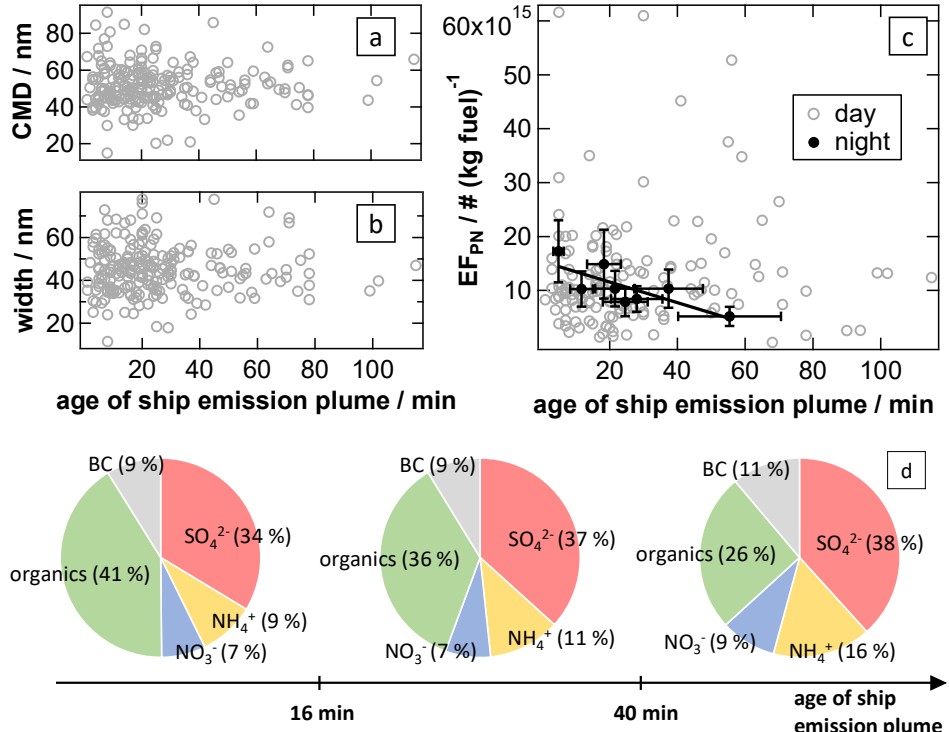

**Figure 9: Influences of atmospheric processing on the particle phase characteristics of ship emission plumes. The CMD (rel. uncertainty: 27 %) (a) and width (rel. uncertainty: 27 %) (b) of the particle number size distribution and the particle number emission factor (rel. uncertainty: 26 %) (c) are plotted against the age of the ship emission plume (avg. rel. uncertainty: 20 %). The**
5 **fit parameters and correlation coefficient are listed in Table 2. Additionally, the average chemical composition of the particle phase is given for ship emission plumes younger than 16 min, between 16 and 40 min of age and older than 40 min (d).**

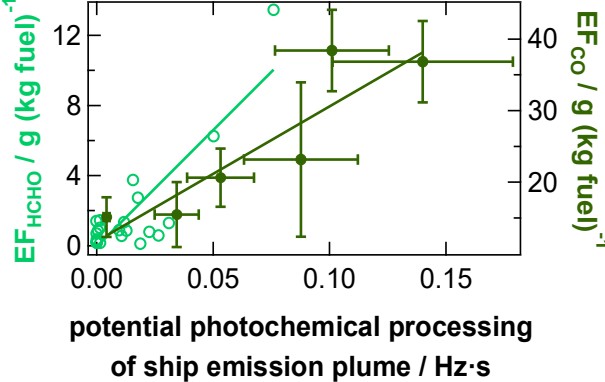

**Figure 10: Influence of atmospheric processing on HCHO (rel. uncertainty: 29 %) and CO in ship emission plumes observed**
10 **during AQABA. For both cases, the development of the emission factor with increasing potential photochemical plume processing (avg. rel. uncertainty: 27 %; measured/calculated $J O^1 D$ integrated over the plume transport time) is presented. The fit parameters and correlation coefficients are listed in Table 2.**





**Figure 11: Qualitative overview of the observed dependencies of ship plume characteristics on ship parameters and conditions during the combustion process as well as on atmospheric processing immediately after plume release and during plume aging.**