# Peer review of "Influence of vessel characteristics and atmospheric processes on the gas and particle phase of ship emission plumes: In–situ measurements in the Mediterranean Sea and around the Arabian Peninsula"

_Atmospheric Chemistry and Physics, 2019_

## Referee Comment (RC1) · Anonymous Referee #1 · 24 Jan 2020

**Review of** *"Influence of vessel characteristics and atmospheric processes on the gas and particle phase of ship emission plumes: In–situ measurements in the Mediterranean Sea and around the Arabian Peninsula"* **by Celik et al, submitted to ACPD.**

In their manuscript, the authors present data and analysis of measurements of over 250 ship emission plumes in the Mediterranean and around the Arabian Peninsula. From the data, the authors present information on the dispersion of the ship plumes, as well as comprehensive data on different components of the emission plumes. The manuscript also connects these observations to ship characteristics obtained from the AIS system. Finally, the authors also discuss the different atmospheric processes that affect the various components of the plume.

The manuscript's topic is very well suited to Atmospheric Chemistry and Physics, and the data and its analysis are interesting and of potentially high value. The paper is well structured and easy to read. Therefore, in principle the paper should be considered for publication in ACP. However, I think there are a few mistakes in the methodology and the reasoning behind drawing some conclusions, especially with a) the interpretation of the black carbon (BC) data and b) the computation of the O:C ratios. These are significant enough that they must be corrected before the paper is published in ACP, as they might change some of the presented results.

In the following, I will first discuss these two possible issues, and then give other general comments on the manuscript:

1. **BC data dependence on pressure and temperature**

I have serious reservations regarding the conclusion driven from the BC data dependence on ambient temperature and pressure (if I understood correctly). To my understanding, the ambient temperature and pressure only affect the gas concentration (in units of molecules/m$^3$) and don't affect the fraction of oxygen in a given mass of air. As in a diesel engine the air is in any case highly compressed, the amount of oxygen available should not change too much. There certainly is the possibility that there might be other pressure/temperature effects that affect engine operation, but to use p/T as a proxy for these, the effects should be more thoroughly explained and appropriate references given.

Additionally, I see a potential alternative explanation to the observation: I'm not fully aware on how the flow calibration for the BC instrument has been performed. However, If the instrument pump pushes a constant mass (or constant number of molecules) per unit time through the filter, i.e. $n = pV/kT$ = constant (as would be the case with a mass flow controller), then as the value of p/T increases, the volume going through the filter decreases. As the derived BC concentration depends in a non-linear way (as explained in Drinovec et al., 2015, which the authors also cite) from the change in attenuation and also the volume flow rate, I think it is conceivable that the observation in Fig 3 is caused by this effect. This should be considered seriously, especially with the reported periodicity of the instrument reading with respect to the measurement container temperature control.

I would suggest reviewing and re-analysing the BC data and its conclusions. For example, it would be informative to see if the BC concentration vs. p/T -dependence is visible only for plumes, or also for ambient BC concentration measurements.

**2.    Calculation of O:C ratios**

I had some trouble following the definitios in eqs 3-5; I think there's an error in the definition of the terms.  Assuming that O/C and H/C ratios refer to the atomic ratios, the second version for the computation of the concentration of [O] and [C] are wrong. This can be illustrated with an example: an organic compound with 5 oxygens, 3 hydrogens, and 3 carbons, so we get the following values

| Compound | N | $M_X$ | $m_X = M_X \bullet N$ |
|---|---|---|---|
| O | 5 | 16 | 80 |
| H | 3 | 1 | 3 |
| C | 3 | 12 | 36 |

The total mass (i.e. [organic] is 119, and O/C = 5:3 and H:C = 3:3.

From this example, we would get

$$[O] = [\text{organic}] \frac{m_O}{m_O + m_H + m_C} = [119] \frac{80}{80 + 3 + 36} = [119] \frac{80}{119} = 80.$$

According to eq. (3), this should be equal to

$$[\text{organic}] \frac{\frac{O}{C} \cdot M_O}{\frac{O}{C} \cdot M_O + \frac{H}{C} \cdot M_H + \frac{C}{C} \cdot M_C} = [119] \frac{\frac{5}{3} 80}{\frac{5}{3} 80 + \frac{3}{3} 3 + \frac{3}{3} 36} = [119] \frac{5 \cdot 80}{5 \cdot 80 + 3 + 36}$$

$$= [119] \frac{400}{439} = 108.43$$

which gives an error of 35% compared to the first equation for [O].

For these calculations, it is also unclear, which of the equations was used for the actual computation of the values. This should be clarified, and in case it has an effect on the results, the new results presented in the revised manuscript.

**3. General Comments**

- Pg.2, line 1: "The results enable identification of...": I think the identification has already been done, so this should be reflected here.
- Pg 4, line 14: "...appropriate inlet systems": The sampling lines can have profound effects on the measured quantities, so I think it would be good to elaborate a little on the sampling line system. If some known guidelines for the sampling were followed, a reference could be given. Are the losses given in Table 1 the line losses? If yes, were they calculated or experimentally determined?

- Page 5, line 15: I think it could be good to clarify the PM1 calculation algorithm: I think based on the explanation it is
$$PM1 = V_{\text{FMPS}} * \rho_{\text{AMS+BC}} * F_{\text{corr}}$$

where $V_{FMPS}$ is the volume obtained from FMPS, $\rho_{AMS+BC}$ is the average density obtained from AMS and BC measurements, and $F_{corr}$ is a correction factor that mostly accounts for underestimated concentrations >130 nm. As the correction factors are averages estimated from the data, I think it would be good to give some information on the variation, eg. give the standard deviation of $\rho_{AMS+BC}$ and $F_{corr}$. Also, if the OPC was measuring large particles, should it be included in Table 1?

- Page 5, line 31: "..linear or Gaussian fits." Here it is unclear what was fitted to what. Could this be elaborated?
- Page 8, line 11: "..linearly interpolated": I think this is just taking the mean background from before and after the plume. Is this so?
- Line 14-15: "defined as …": it would be easier to read this just as a formula I think.
- Page 9, line 4: "dispersion time constant": As there are several ways of defining time constants, I think it is important to say which one is meant. Is it the concentration half-life, or e-folding time, or some other one?
- Page 9, line 13: "one standard deviation": I think this should be geometric standard deviation
- Page 9, line 20: "measured OH concentrations were not used… only describe the situation and the research vessel's position and and not within the plume". Is this not also the case for the photolysis rate?
- Page 12, line 5-10: A reduction of particle number EF was observed with vessel size, and this is attributed to coagulation. Is this the only explanation, or could there be others too? What is the variation in the residence time?
- Page 12, line 12, and pg 19, line 7. -> I find the finding on the wind speed influencing the effective EF interesting. I think this is an important effect to take into consideration. However, I think that it should be made clear that at the source, the emission is the same – the difference in the observed EF is introduced only later during the processing of the plume. Additionally, I'd like to know the more detailed reasoning behind this, as to me, the increased dilution in at high WS should be accounted for in the calculation. Is the age of the plume (since emission) factored in here? Does the wind speed affect engine operation?
- With regard to the wind speed, was there a correlation with the plume dispersion time constant (page 9) and wind speed? As the explanation for the effect on particles is related to dilution, I think there might be a connection between the two. If not, why?
- More generally, I think that the term emission factor (EF) should be used only for actual emission at the point and time of emission, and if the observed aerosol after some time has some differing characteristics, this should be noted with some qualifier, e.g. 'apparent' or 'effective'.
- Figure 2: I have some trouble understanding the decision-making when two ships are in the path of an air mass. In the example given, it seems to me from the figure that both the red and the blue ship are relevant for the observed plume. Why was the blue ship not considered? This could be clarified.

---

## Referee Comment (RC2) · Mingxi Yang (Referee) · 11 Feb 2020

This paper presents an analysis of a large dataset of ship plumes measured from a research vessel in the Mediterranean. It's a very impressive data set with many gas and particle phase instruments. The authors used AIS ship positioning to attribute measured ship plumes to individual ships, a non-trivial exercise. The various emission factors were computed (accounting for plume dilution) and then related to

ship/engine characteristics, operational conditions, atmospheric conditions, etc. ÂăThe paper is generally well written and the data will be very useful for better understanding of ship emissions.

Because a large number of ships were measured, and because it's not possible to follow individual ship plumes in a Lagrangian fashion from a research vessel, the analysis is generally done on all the detected plumes as a whole, with a lot of bin-averaging involved.Âă This unfortunately buries many details and real sources of variability.Âă I would like to see the authors trying to tease out some of those complexities (see detailed comments below).Âă I think the authors should make it clear at the beginning that their measurements are not Lagrangian (i.e. following the same ship plume over time).Âă Thus their observations of the effect of plume aging represent the variability of an ensemble of initially different plumes at various stages of aging.Âă The observations don't uniquely capture the plume aging effect and this should be acknowledged.Âă

The different emission factors as a function of plume age/photochemical processing is very interesting.Âă Can author say something about the max distance (or time) down wind of source for which the various EF estimates still represent stack emissions?

I wonder whether it'd be useful to have two different terminologies: 1) emission factors for very close to the emitting ship, representing what's coming out of the stack (=emission inventories for modelling); 2) x ratios (or named something else) that represents 'emission factors' further downwind.Âă This would make it clearer that the 'emission factors' observed from a ship far away do not always represent what's initially coming out of the stack.

Some specific comments:

plume age 20 min. transported over 4 km. implies very low winds in general low winds (poor dispersion) contributes to $CO_2$ being detectable for so long? p. 9.Âă evidence of enhanced OH concentrations within the plumes? Fig.3aÂă given the small range in ambient $O_2$ concentration (at most a few %), I'm surprised that the BC EF changes so

much.Âă does humidity affect the BC EF?Fig.3bÂă the NO:NO2 ratio depends on the reactions with O3 and so plume age.Âă it's also probably affected by photochemistry. those complexities are not teased out in this figure.Âă perhaps limit the analysis only to a certain plume age?

Fig3. Does organic EF vary with combustion efficiency or ship velocity? Fig.5 as eluded to in section 3.2 the importance of coagulation on aerosol number emission also depend on the plume age to some extent. that complexity is again not teased out here Fig.6 besides coagulation and condensation, could wind speed or sea state alter the performances of the ships engines, and hence the aerosol emission factors?

P 13 line 16. 1/3 of initial ratio? Initially it's almost all NO

SO2/sO4 ratio decreases with humidity. Could this be partly due to cloud processing (if cloudy)?

Fig9 changes in aerosol composition with aging. Would be interesting to see this separated to daytime and nighttime

Relationship between O/C ratio and organic EF. Unclear whether 'increase in mass through Oxidation' occurs in atmosphere or in stack

P 18 line 19. Higher NO:NO2?

---

## Author Comment (AC1) · 17 Mar 2020

**Reply to the reviewers.**

**Review 1:**

**Review of *"Influence of vessel characteristics and atmospheric processes on the gas and particle phase of ship emission plumes: In–situ measurements in the Mediterranean Sea and around the Arabian Peninsula"* by Celik et al, submitted to ACPD.**

In their manuscript, the authors present data and analysis of measurements of over 250 ship emission plumes in the Mediterranean and around the Arabian Peninsula. From the data, the authors present information on the dispersion of the ship plumes, as well as comprehensive data on different components of the emission plumes. The manuscript also connects these observations to ship characteristics obtained from the AIS system. Finally, the authors also discuss the different atmospheric processes that affect the various components of the plume.

The manuscript's topic is very well suited to Atmospheric Chemistry and Physics, and the data and its analysis are interesting and of potentially high value. The paper is well structured and easy to read. Therefore, in principle the paper should be considered for publication in ACP. However, I think there are a few mistakes in the methodology and the reasoning behind drawing some conclusions, especially with a) the interpretation of the black carbon (BC) data and b) the computation of the O:C ratios. These are significant enough that they must be corrected before the paper is published in ACP, as they might change some of the presented results.

*Reply: We thank the reviewer for this very positive general comment on our manuscript.*

In the following, I will first discuss these two possible issues, and then give other general comments on the manuscript:

**1. BC data dependence on pressure and temperature**
I have serious reservations regarding the conclusion driven from the BC data dependence on ambient temperature and pressure (if I understood correctly). To my understanding, the ambient temperature and pressure only affect the gas concentration (in units of molecules/m3) and don't affect the fraction of oxygen in a given mass of air. As in a diesel engine the air is in any case highly compressed, the amount of oxygen available should not change too much. There certainly is the possibility that there might be other pressure/temperature effects that affect engine operation, but to use p/T as a proxy for these, the effects should be more thoroughly explained and appropriate references given.

Additionally, I see a potential alternative explanation to the observation: I'm not fully aware on how the flow calibration for the BC instrument has been performed. However, If the instrument pump pushes a constant mass (or constant number of molecules) per unit time through the filter, i.e. n = pV/kT = constant (as would be the case with a mass flow controller), then as the value of p/T increases, the volume going through the filter decreases. As the derived BC concentration depends in a non-linear way (as explained in Drinovec et al., 2015, which the authors also cite) from the change in attenuation and also the volume flow rate, I think it is conceivable that the observation in Fig 3 is caused by this effect. This should

be considered seriously, especially with the reported periodicity of the instrument reading with respect to the measurement container temperature control.

I would suggest reviewing and re-analysing the BC data and its conclusions. For example, it would be informative to see if the BC concentration vs. p/T -dependence is visible only for plumes, or also for ambient BC concentration measurements.

*Reply: Thank you very much for this very important comment. As a consequence of this comment we thoroughly reviewed the BC data. According to the manufacturer of the Aethalometer, which was used to determine BC concentrations, the instrument uses a mass flow controller to regulate the flow through the instrument. However, such effects as described by the reviewer should still not occur, because in this instrument the pressure and temperature of the flow are also monitored and the measurement results are internally converted to standard conditions.*
*Unfortunately, the BC concentrations vs. p/T-dependence of ambient BC concentration measurements cannot be assessed, since in ambient measurements BC concentrations depend strongly on the air mass history.*

*In order to assess this issue further we have determined whether we can also find a p/T dependency of the NOx emission factor, which is also known to have a dependency on combustion efficiency. In agreement with our findings for BC we found an increase in $EF_{NOx}$ with increasing p/T, i.e. also suggesting increasing combustion efficiency with increasing p/T. We find it very unlikely that both of these independent instruments generate inverted trends (BC decreases, NOx increases) as a consequence of p/T changes.*

*In a further step we have searched the literature for information on internal combustion engine performance dependency on ambient air conditions. According to the literature a decrease in ambient pressure and an increase in ambient temperature can result in reduced engine performance, in agreement with our findings (Bermudez et al., 2017; Chang et al., 2017; Rajewski, 2018). Furthermore, ambient humidity is also known to deteriorate combustion efficiency, resulting in reduced NOx and increased soot emissions. In agreement with this, we also found a general increase in our BC emission factors and a decrease in our NOx emission factors with increasing ambient (absolute) humidity. The use of turbochargers and intercoolers in marine diesel engines reduces the influence of ambient conditions onto the combustion process, but apparently does not completely suppress it. This is also in accordance with information from the literature (Chang et al., 2017).*

*As a consequence of these findings we slightly extended the respective statement and added the correlation plots for BC on H2O concentration as well as for NOx on p/T and H2O concentration to the manuscript (see revised Fig. 3). The statement in the main text now reads:*

"… In general, more efficient fuel combustion results in increased $NO_x$ (especially NO) emissions owing to the enhanced oxidation of atmospheric nitrogen that occurs at higher combustion temperatures. It also leads to decreased soot particle (*here*: BC) emissions as a consequence of more efficient oxidation of fuel carbon (Corbett et al., 1999; Juwono et al., 2013; Pokhrel and Lee, 2015). The fuel combustion efficiency depends primarily on the oxygen–to–fuel mixing ratio in the combustion chamber (Khalid, 2013), but also increased ambient temperature and absolute humidity as well as reduced ambient pressure can result in deteriorated combustion efficiencies of diesel

engines, associated with elevated soot and reduced NO$_x$ emissions (Bermudez et al., 2017; Chang et al., 2017; Rajewski, 2018). Even though the use of turbochargers and intercoolers in marine diesel engines should minimize the influence of ambient conditions onto the combustion process, we found indications for combustion efficiency dependence on such variables. With increasing ambient pressure *p* and decreasing temperature *T*, as well as for an increase in the p/T ratio (Fig. 3(a)), a reduction of the BC emission factor by more than a factor of 2 over the range of observations and a general increase of the NO$_x$ emission factor were observed, both indications for improved combustion efficiency (see below). Likewise, deteriorated combustion efficiency with increasing ambient absolute humidity is suggested by the observed dependencies of the BC and NO$_x$ emission factors on ambient water vapor concentration (Fig. 3(b)). "

**2. Calculation of O:C ratios**

I had some trouble following the definitions in eqs 3-5; I think there's an error in the definition of the terms. Assuming that O/C and H/C ratios refer to the atomic ratios, the second version for the computation of the concentration of [O] and [C] are wrong. This can be illustrated with an example: an organic compound with 5 oxygens, 3 hydrogens, and 3 carbons, so we get the following values

| Compound | N | M$_x$ | m$_x$=M$_x$•N |
|----------|---|-------|----------------|
| O | 5 | 16 | 80 |
| H | 3 | 1 | 3 |
| C | 3 | 12 | 36 |

The total mass (i.e. [organic] is 119, and O/C = 5:3 and H:C = 3:3.

From this example, we would get

$$[O] = [\text{organic}] \frac{m_O}{m_O + m_H + m_C} = [119] \frac{80}{80+3+36} = [119] \frac{80}{119} = 80.$$

According to eq. (3), this should be equal to

$$[\text{organic}] \frac{\frac{O}{C} \cdot M_O}{\frac{O}{C} \cdot M_O + \frac{H}{C} \cdot M_H + \frac{C}{C} \cdot M_C} = [119] \frac{\frac{5}{3} 80}{\frac{5}{3} 80 + \frac{3}{3} 3 + \frac{3}{3} 36} = [119] \frac{5 \cdot 80}{5 \cdot 80 + 3 + 36}$$

$$= [119] \frac{400}{439} = 108.43$$

which gives an error of 35% compared to the first equation for [O].

*Reply: We have checked the formula once again and did not find an error with it. In the calculation presented by the reviewer, however, we found an error:*

*Using the values from the table provided by the reviewer (C3H3O5 as example substance) in the formula for [O] we get, largely in agreement with the calculation by the reviewer*

$$[O] = [organic] \frac{m_O}{m_O + m_H + m_C} = [organic] \frac{80}{80 + 3 + 36} = [organic] * \frac{80}{119}$$

*Note, that [organic] is the actually measured total particulate organics mass concentration, determined by the AMS. It is not the molecular weight of an individual molecule. In the revised version of the manuscript we stress this further.*

*According to equation (3) this should be equal to*

$$[organic] \frac{\frac{O}{C} M_O}{\frac{O}{C} M_O + \frac{H}{C} M_H + \frac{C}{C} M_C} = [organic] \frac{\frac{5}{3} 16}{\frac{5}{3} 16 + \frac{3}{3} 1 + \frac{3}{3} 12}$$

$$= [organic] \frac{80}{80 + 3 + 36} = [organic] * \frac{80}{119}$$

*The difference (besides the fact that for [organic] no value was introduced, which however does not make a difference in the overall calculation) is that for $M_O$, $M_H$, and $M_C$ the atomic weights of O, H, and C need to be used (as stated in the manuscript text and in the table above) and not their atomic weights multiplied by the number of atoms in the molecule. Then the result of both formulae is the same as presented above.*

*To avoid confusion, we therefore slightly extended the text around the formula to make this clearer:*

"Using the measured mass concentrations of particulate organics ([organics]) and the atomic O/C and H/C ratios for the organic aerosol during plume and background measurements, the O/C ratios for the plume contribution were calculated (Eq. 6) from the average mass concentrations of oxygen and carbon for both the background (B) and for the ship emission event (emission + background, EB). The mass concentrations of O and C were calculated from the mass fraction of oxygen and carbon in the organic aerosol, determined from the measured atomic O/C and H/C ratios, following Eqs. (4) and (5):

$$[O] = \frac{[organics] \cdot m_O}{m_O + m_H + m_C} = \frac{[organics] \cdot O/C \cdot M_O}{O/C \cdot M_O + H/C \cdot M_H + C/C \cdot M_C}, \tag{4}$$

$$[C] = \frac{[organics] \cdot m_C}{m_O + m_H + m_C} = \frac{[organics] \cdot M_C}{O/C \cdot M_O + H/C \cdot M_H + M_C}, \tag{5}$$

$$O/C = \frac{([O]_{EB} - [O]_B)/M_O}{([C]_{EB} - [C]_B)/M_C}, \tag{6}$$

where $m_x$ is the (measured) total mass and $M_x$ the atomic weight of the respective species $x$. For these calculations we assume that particulate organics consist only of oxygen, hydrogen and carbon. To check for consistency, the H/C ratio was calculated analogously and found to always show the expected reverse behaviour to the O/C ratio."

For these calculations, it is also unclear, which of the equations was used for the actual computation of the values. This should be clarified, and in case it has an effect on the results, the new results presented in the revised manuscript.

*Reply: As discussed above, both equations are equal. The second equation was used to calculate [O] and [C]. To make this clearer we also added some text to this equation as shown above.*

**3. General Comments**
- Pg.2, line 1: "The results enable identification of…": I think the identification has already been done, so this should be reflected here.

*Reply: Thank you for this comment. The sentence was not clear enough and could be misunderstood. Therefore, we changed it to: "The results allow to describe the influences on (or processes in) ship emission plumes quantitatively by parameterizations, which could be used for further refinement of atmospheric models, and to identify which of these processes are the most important ones."*

- Pg 4, line 14: "…appropriate inlet systems": The sampling lines can have profound effects on the measured quantities, so I think it would be good to elaborate a little on the sampling line system. If some known guidelines for the sampling were followed, a reference could be given. Are the losses given in Table 1 the line losses? If yes, were they calculated or experimentally determined?

*Reply: Within this study, measurements from 11 different instruments were used, which utilized a variety of different inlet lines, each optimized for the respective measurement. For the aerosol instruments a common inlet system was used. Particle losses within this inlet system were estimated using the Particle Loss Calculator (von der Weiden, et al., 2009) as mentioned in the main text. We now added this information also to the caption of Table 1 and also made clearer that these losses are transport losses. To avoid a lengthy discussion on the various inlet systems of all instruments we need to refer to the respective publications where the various data sets are presented by the operators of the instruments for more details.*

- Page 5, line 15: I think it could be good to clarify the PM1 calculation algorithm: I think based on the explanation it is PM1 = $V_{FMPS}$ * $r_{AMS+BC}$*$F_{corr}$
where $V_{FMPS}$ is the volume obtained from FMPS, $r_{AMS+BC}$ is the average density obtained from AMS and BC measurements, and $F_{corr}$ is a correction factor that mostly accounts for underestimated concentrations >130 nm. As the correction factors are averages estimated from the data, I think it would be good to give some information on the variation, eg. Give the standard deviation of $r_{AMS+BC}$ and $F_{corr}$. Also, if the OPC was measuring large particles, should it be included in Table 1?

*Reply: Thank you for this suggestion. To improve clarity of the PM1 calculation procedure we included the suggested formula into the text and added the other requested information:*
*"As the ship emission plumes measured during the AQABA field campaign showed particles exclusively in the (lower) size range of the FMPS, this data was used to calculate $PM_1$ particle mass concentrations from the FMPS-derived total particle volume concentration ($V_{FMPS}$), assuming spherical particles and an average particle density of 1.53 g cm$^{-3}$ calculated using the mass concentrations of AMS species and BC ($\rho_{AMS+BC}$). The calculated $PM_1$ particle mass concentrations were corrected for under–measurement in the upper size range of the FMPS (> 130 nm; Levin et al., 2015) by scaling them with a factor of $F_{corr}$=1.85, which was derived from comparison with size distribution data from a concurrently measuring Optical Particle Counter (OPC, Grimm Model 1.109):*

$$PM_1 = V_{FMPS} \cdot \rho_{AMS+BC} \cdot F_{corr} \quad . \tag{1}$$

While $V_{FMPS}$ are actually measured data for each plume event, $F_{corr}$ and $\rho_{AMS+BC}$ are the average correction factor for the under-measurement in the upper FMPS channels and the average particle density, determined from AMS and BC measurements for the whole field campaign. The overall uncertainty of the resulting $PM_1$ concentrations is estimated to be 35%."

*As mentioned in the manuscript, no large particles (larger than the size range covered by the FMPS) were found in the ship emission plumes. Therefore, the OPC data were not used to derive information on the ship emissions. The OPC data from the out-of-plume measurements were only used to correct the upper size range of the FMPS for under-measurement as mentioned in the text. Because the OPC data are not included in the presented analysis, we feel that this instrument should not be included in Table 1, as well as many other instruments which were operated on-board the research vessel and which were also not used for this analysis.*

- Page 5, line 31: "..linear or Gaussian fits." Here it is unclear what was fitted to what. Could this be elaborated?

*Reply: We re-worded the corresponding sentences to make clearer what we have done:*
"Some of the gas phase data, especially those of the nitrogen oxides, show periodic gaps of 1 to 2 min duration due to periodic background measurements or calibrations. These affected in some cases the detected ship emission events. If possible, the affected events were reconstructed using the remaining parts of the times series by either linear or Gaussian fits, depending on the expected shape of the missing part of the peak."

- Page 8, line 11: "..linearly interpolated": I think this is just taking the mean background from before and after the plume. Is this so?

*Reply: This is correct. We re-worded this sentence to make this clear:*
"The mean background was calculated from the average values derived for the two background intervals and subtracted from the average event concentration to obtain the average excess concentration in the plume."

- Line 14-15: "defined as …": it would be easier to read this just as a formula I think.

*Reply: We agree with the reviewer and expressed the definition of the detection limit as formula:*
"In general, calculated average excess concentrations below the detection limit ($LOD = \frac{3\,\sigma_{bg}}{\sqrt{n}}$ , with $\sigma_{bg}$ the standard deviation of the background and *n* the number of measurement points within the ship emission event interval) were excluded from further analysis."

- Page 9, line 4: "dispersion time constant": As there are several ways of defining time constants, I think it is important to say which one is meant. Is it the concentration half-life, or e-folding time, or some other one?

*Reply: As dispersion time constant we use the e-folding time. We added this information to the text.*

- Page 9, line 13: "one standard deviation": I think this should be geometric standard deviation

*Reply: Thank you. We added this to the text.*

- Page 9, line 20: "measured OH concentrations were not used… only describe the situation and the research vessel's position and not within the plume". Is this not also the case for the photolysis rate?

*Reply: The photolysis rate $JO^1D$ is determined from the spectrally resolved irradiation. Only under special conditions, e.g. when there are strong spatial gradients in cloud coverage, this variable is strongly different along the path from the emission position to the measurement position. Generally, we assume that it is a much more robust variable compared to the locally measured OH concentration, which is affected by a multitude of influences. Therefore, we assume that normally this variable is a good proxy for potential photochemical processing during the plume transport. To make this point clearer we changed the sentence to:*
"Measured OH concentrations were not used as a measure of photochemical processing due to insufficient data coverage and because they are more affected by local influences than the photolysis rate and therefore more so than the latter describe merely the situation at the research vessel's position rather than within the plume."

- Page 12, line 5-10: A reduction of particle number EF was observed with vessel size, and this is attributed to coagulation. Is this the only explanation, or could there be others too? What is the variation in the residence time?

*Reply: Of course, there are many potential reasons why particle number emission factors could vary. This includes (unknown) details about the design of the burning chambers, the burning conditions or other factors which affect the generation of particulate emissions. Typically, an important factor influencing particle number concentrations is coagulation at high particle number concentrations. Since we expect longer residence times in the exhaust system of larger ships, it makes sense to assume a stronger influence of coagulation on particle number concentration in such ships. Since we state that our observation is "in agreement with this", we do not claim that this is the one and only explanation for our observation. We do not know the residence time of the exhaust in the individual ships and therefore we also do not know the variation of residence times. We therefore clarified this in this sentence:*
"Longer residence times within the exhaust system, as they typically occur in larger ships, should therefore result in lower particle number concentrations due to enhanced coagulation effects."

- Page 12, line 12, and pg 19, line 7. -> I find the finding on the wind speed influencing the effective EF interesting. I think this is an important effect to take into consideration. However, I think that it should be made clear that at the source, the emission is the same – the difference in the observed EF is introduced only later during the processing of the plume. Additionally, I'd like to know the more detailed reasoning behind this, as to me, the increased dilution in at high WS should be accounted for in the calculation. Is the age of the plume (since emission) factored in here? Does the wind speed affect engine operation?

*Reply: As written within the text, we treat the initial dilution of the exhaust from the stack of the ship into the atmosphere as part of the emission process. However, we agree with the reviewer that we should make the influences of the different parts of this emission process clearer. Therefore, we changed the sentence to:*
*"However, even for a given particle number concentration at the location of emission, the level of further coagulation depends on the concentration level in the transported plume, which is expected to depend on the ambient wind speed, as it influences the degree of dilution in this phase of emission."*
*Since we do not discuss absolute concentrations but emission factors (i.e. concentrations normalized to defined amounts of fuel), the effects of dilution (both, due to different wind speed as well as due to different transport times) are already accounted for in the EFs. Therefore, if we find an influence of wind speed onto particle number concentration EFs, this should reflect a real influence of wind speed onto particle number concentrations in the emission plumes.*
*Because on average ships do not travel in a certain direction with respect to wind direction (e.g. preferentially in opposite direction as the wind direction), we do not expect a general influence of wind speed on engine operation.*

- With regard to the wind speed, was there a correlation with the plume dispersion time constant (page 9) and wind speed? As the explanation for the effect on particles is related to dilution, I think there might be a connection between the two. If not, why?

*Reply: This is an interesting point. Unfortunately, due to the many factors influencing the various emission factors, concentrations and other variables, it was not possible to determine e.g. dispersion time constants separately for different wind speed conditions. Therefore, we unfortunately cannot answer this question.*

- More generally, I think that the term emission factor (EF) should be used only for actual emission at the point and time of emission, and if the observed aerosol after some time has some differing characteristics, this should be noted with some qualifier, e.g. 'apparent' or 'effective'.

*Reply: Thank you, this is a very good point. We revised the usage throughout the text and use the term "emission factor" only for those EF which were either determined for the point of emission or do not change over transport time, and use the term "apparent emission factor" for those EF which were altered during processes occurring during transport. We introduce*

*the term "apparent emission factor" in Sect. 2.4, where the general emission factor calculation is presented (page 9).*

- Figure 2: I have some trouble understanding the decision-making when two ships are in the path of an air mass. In the example given, it seems to me from the figure that both the red and the blue ship are relevant for the observed plume. Why was the blue ship not considered? This could be clarified.

*Reply: The position of the ships at the time when the back-traced air mass intersects the reconstructed ship track is indicated by the "ship" label (see legend of Figure 2). For the red ship this position is within the wind direction sector, while for the blue ship it is outside this sector. Therefore, at the time when the air mass intersected the ship track, the red ship was at the intersection point while the blue one was not.*
*To make this clearer we re-worded the figure caption:*
*"Two AIS records (position indicated by dots on the vessel tracks) were requested after the AQABA field campaign from the AIS data base for each of these vessels … The uncertainties of the intersection times and of the corresponding vessel positions (ship markers on the vessel tracks) are based on the uncertainties of the wind speed."*

References:

Bermudez, V., Serrano, J. R., Piqueras, P., Gomez, J., Bender, S.: Analysis of the role of altitude on diesel engine performance and emissions using an atmosphere simulator, International Journal of Engine Research 18, 105-117, DOI:10.1177/1468087416679569, 2017.

Chang, Y., Mendrea, B., Sterniak, J., Bohac, S. V.: Effect of Ambient Temperature and Humidity on Combustion and Emissions of a Spark-Assisted Compression Ignition Engine, J. Eng. Gas Turbines Power, 139, 51501-1 – 51501-7, https://doi.org/10.1115/1.4034966, 2017.

Rajewski, A.: Evaluating internal combustion engine's performance, Wärtsilä Technical Journal 2/2018, 18-23, 2018.

**Review 2:**
This paper presents an analysis of a large dataset of ship plumes measured from a research vessel in the Mediterranean. It's a very impressive data set with many gas and particle phase instruments. The

authors used AIS ship positioning to attribute measured ship plumes to individual ships, a non-trivial exercise.

The various emission factors were computed (accounting for plume dilution) and then related to ship/engine characteristics, operational conditions, atmospheric conditions, etc. The paper is generally well written and the data will be very useful for better understanding of ship emissions.

*Reply: We thank the reviewer for this very positive overall comment on our manuscript.*

Because a large number of ships were measured, and because it's not possible to follow individual ship plumes in a Lagrangian fashion from a research vessel, the analysis is generally done on all the detected plumes as a whole, with a lot of bin-averaging involved. This unfortunately buries many details and real sources of variability. I would like to see the authors trying to tease out some of those complexities (see detailed comments below). I think the authors should make it clear at the beginning that their measurements are not Lagrangian (i.e. following the same ship plume over time). Thus, their observations of the effect of plume aging represent the variability of an ensemble of initially different plumes at various stages of aging. The observations don't uniquely capture the plume aging effect and this should be acknowledged.

*Reply: Thank you for this important comment. We realize that we did not make sufficiently clear that our analysis approach is not based on Lagrangian measurements of individual plumes but on the measurement of plume ensembles and deduction of dependencies on various variables from such ensembles. We revised the paragraph where such general remarks on our analysis approach are introduced (beginning of Sect. 3; see below) to make this clearer.*

*The reviewer is right, bin-averaging does bury much of the variation within the measurement data. On the other hand, it provides a clearer picture to the reader about potential dependencies of emission factors on certain external variables. To make use of both, the whole information within the un-binned data and the improved clarity of the binned emission factors, we used the former ones to calculate the correlations and parameterizations provided in Table 2 and the latter ones to generate most of the Figures. To make this clearer, we revised the respective text in the manuscript (Sect. 3; see below).*

*We completely agree with the reviewer that within the variability of the un-binned data a lot of information about additional dependencies of emission factors on other external parameters is buried. In our analysis of the data we tried to tease out this information with three different methods, (1) color-coding the markers in the correlation plots depending on the magnitude of a third variable, (2) producing separate correlation plots for sub-sets of the emission factors, and (3) correlating the emissions factors with several potentially influencing factors in separate correlation plots. Because most of the emission factors depend on multiple (and not only one or two) external parameters and because the number of data points in each of such plots is very limited (i.e. to the number of available data points, i.e. plumes, for which all variables of interest are available at the same time), the first two of these approaches did not result in conclusive additional information for most of the emission factors. Only for a few of the correlations significant differences were found (and are already presented in the manuscript), e.g. in some cases when differentiating between day and night measurements, as for example done for the particle number emission factor (Figure 9c). For most of the emission factors where differences between night and day were found, it was more conclusive to use photochemical plume age as independent variable in the correlation plots, which already includes this effect (day/night). For individual emission factors mentioned in the detailed comments below we provide the respective correlation plots with separated correlations for day and night either in this reply or in the supplementary material (see replies to comments below). In addition, we add this information about the reason why no additional secondary dependencies of emission factors could be teased out and therefore are not presented in the revised manuscript (Sect. 3):*

"Parallel measurements of multiple variables which are associated with ship emission plumes and observation of such plumes under very different conditions (e.g. plume age, meteorological conditions, source vessel characteristics, etc.) allow the investigation of factors which might influence the characteristics of ship emission plumes. For this purpose, we investigated the relationship between plume characteristics and various factors (above mentioned measurement conditions) by correlation analysis, which, contrary to Lagrangian measurements of individual ship plumes during transport away from the source, provide ensemble-averaged information on emission or transformation characteristics of plume properties. As several different factors can influence individual plume characteristics, the correlation plots always show a relatively strong degree of scatter, which led us to bin them, such that the same number of data points (at least 5 and maximal 32) was included in each bin, resulting in not equidistantly distributed bins. The slope and intercept did not significantly change when using binned data instead of raw data. Therefore, fit parameters for the correlation of raw data are presented in Table 2, while in the figures usually the binned data are shown for clarity. In case raw data are presented their relative uncertainties combine the estimated quantification (Sect. 2.4) and measurement uncertainties, whereas in case of binned data error bars include in addition one sigma standard deviations of the data distributions in each bin. Using bin-averaged data or linear regressions strongly reduces the influence of additional influencing factors on the individual data points and, to a large degree, provides information on the dependence of the respective plume feature on the influencing factor under investigation. To extract the influence of several influencing external factors on plume characteristics, separate correlation plots for each of these factors were therefore generated."

The different emission factors as a function of plume age/photochemical processing is very interesting. Can author say something about the max distance (or time) down wind of source for which the various EF estimates still represent stack emissions?

*Reply: This is a very good point. Indeed, the emission factors determined further downwind of the source do not necessarily represent actual stack emissions. This is always not the case when emission factors show a significant dependency on transport time or distance. To make this clear and also in agreement with the following comment we introduced the term "apparent emission factor" for those EFs which undergo changes during transport and therefore do not reflect the situation at the point of emission. Those EFs which do not undergo such changes during transport are termed "emission factor" in the revised manuscript. Please see also the reply to the following comment below for further details.*

I wonder whether it'd be useful to have two different terminologies: 1) emission factors for very close to the emitting ship, representing what's coming out of the stack (=emission inventories for modelling); 2) x ratios (or named something else) that represents 'emission factors' further downwind. This would make it clearer that the 'emission factors' observed from a ship far away do not always represent what's initially coming out of the stack.

*Reply: We completely agree with the reviewer that the usage of two different terminologies for (1) emission factors at the point of emission and (2) emission factors as measured further downwind after atmospheric alteration would provide a clearer picture of what they really mean. Since no measurements directly at the stack of a ship have been performed we distinguish between emission factors for which we do not see a change over time after emission and which therefore can be expected to be the same at the time and point of emission as at the time when we measured them, and emission factors for which we observe a temporal change during transport of the plume. In the revised version of the manuscript we now call the former ones "emission factors" and the latter ones "apparent emission factors" throughout the text. This differentiation is introduced in section 2.4:*

"In this study, emission factors are also used for the investigation of atmospheric processes (except plume dilution) and aging, which result in apparent emission factors at the point of measurement."

Some specific comments:

plume age 20 min. transported over 4 km. implies very low winds in general low winds (poor dispersion) contributes to CO2 being detectable for so long?

*Reply: Indeed, most of the ship emission plumes were observed during conditions with relatively low to moderate wind speed (2 – 6 m/s, i.e. 7 – 21 km/h). However, we do not observe less efficient plume detection with increasing wind speed for plumes which were emitted at larger distance. Therefore, we do not expect a large influence of wind speed on the capability to detect a plume that was emitted at a certain distance from the measurement location. From our data, unfortunately, we cannot provide any information on the dependence of plume dispersion on wind speed.*

p. 9. Evidence of enhanced OH concentrations within the plumes?

*Reply: We have looked at the OH concentrations during the times when the plumes were observed in the other instruments' data sets. The concentrations observed during the plume arrival times, in comparison to those before and after these time intervals, do not show a consistent picture. We are in the process of investigating potential influencing factors which could explain the observed behavior. If successful, this could be the focus of a forthcoming publication.*

Fig.3a given the small range in ambient O2 concentration (at most a few %), I'm surprised that the BC EF changes so much. does humidity affect the BC EF?

*Reply: We apologize for the unclear wording in the discussion of Figure 3 which led to this misunderstanding. With "ambient oxygen concentration" we did not mean the relative contribution of oxygen to ambient air, which indeed changes only marginally, but the change in absolute concentration, i.e. number of molecules per liter. The latter one changes with increasing pressure and decreasing temperature, i.e. with increasing p/T. Further investigation of ship engine performance dependencies on ambient conditions have shown that indeed also ambient humidity does affect the combustion efficiency of such engines (see also reply to reviewer #1). To make the first point clearer and to include the information on ambient absolute humidity as requested by the reviewer, we extended Sect. 3.1 and added the EF(NOx) dependency on p/T as well as the EF(BC) and EF(NOx) dependencies on absolute humidity to Figure 3:*

"**Influence of combustion conditions.** The ship emission plumes of the AQABA dataset enabled us to extract information regarding the influence of combustion conditions. In general, more efficient fuel combustion results in increased $NO_x$ (especially NO) emissions owing to the enhanced oxidation of atmospheric nitrogen that occurs at higher combustion temperatures. It also leads to decreased soot particle (*here*: BC) emissions as a consequence of more efficient oxidation of fuel carbon (Corbett et al., 1999; Juwono et al., 2013; Pokhrel and Lee, 2015). The fuel combustion efficiency depends primarily on the oxygen–to–fuel mixing ratio in the combustion chamber (Khalid, 2013), but also increased ambient temperature and humidity as well as reduced ambient pressure can result in deteriorated combustion efficiencies of diesel engines, associated with elevated soot and reduced $NO_x$ emissions (Bermudez et al., 2017; Chang et al., 2017; Rajewski, 2018). Even though the use of

turbochargers and intercoolers in marine diesel engines should minimize the influence of ambient conditions onto the combustion process, we found indications for combustion efficiency dependence on such variables. With increasing ambient pressure $p$ and decreasing temperature $T$, as well as for an increase in the $p/T$ ratio (Fig. 3(a)), a reduction of the BC emission factor by more than a factor of 2 over the range of observations and a general increase of the $NO_x$ emission factor were observed, both indications for improved combustion efficiency. Likewise, deteriorated combustion efficiency with increasing ambient humidity is suggested by the observed dependencies of the BC and $NO_x$ emission factors on ambient water vapor concentration (Fig. 3(b))."

Fig.3b the NO:NO2 ratio depends on the reactions with O3 and so plume age. it's also probably affected by photochemistry. those complexities are not teased out in this figure. perhaps limit the analysis only to a certain plume age?

*Reply: The reviewer is absolutely right: the NO/NO2 ratio depends not only on ship velocity but also on other factors like plume age as shown in Figure 7a. As mentioned above, most of the emission factors and emission ratios depend on a large number of external factors. For the NO/NO2 ratio we tried to tease out this additional dependency by color-coding the markers in the un-binned correlation depending on the plume age, associated with each individual plume event. Due to additional influences on the NO/NO2 ratio, this does not provide a conclusive picture:*

[Figure]

*Limiting the correlation of NO/NO2 versus vessel speed to data from only one plume age would strongly reduce the number of data points. Nevertheless, there would still be a significant amount of scatter in the remaining data points in the correlation due to additional external influences on the NO/NO2 ratio.*
*We think the best approach to tease out the different dependencies of a certain pollutant ratio or emission factor on various influencing parameters in our case is to present several of these dependencies and provide the respective Pearson's R values to indicate which of the dependencies is the strongest one. To stress the fact that NO/NO2 does not only depend on vessel speed but also on plume age, we added a reference to Figure 7a and Section 3.3 to the discussion of Figure 3c (the former Figure 3b). In addition, we used the plume age dependence of the NO/NO2 ratio which we*

*obtained in Section 3.3 and used it to calculate the "initial NO/NO2 ratio", as it would have been observed close to the point of emission, for each plume. For these initial NO/NO2 ratios we also determined the vessel speed dependence (Figure S13) and added this information to Table 2. The fact that both regressions (for the measured ratios and for the calculated initial ratios) show identical slopes shows that binning of data or calculating regressions for all data largely averages out additional influences on the observables. In order to reflect this, we revised the text as follows (Sect. 3.1):*

"In agreement with this, we find a 3–fold increase in the measured NO to $NO_2$ ratio and an almost 3–fold increase in the $NO_x$ emission factor over the range of observed vessel speeds from 0 to ~10 m s$^{-1}$ (see Fig. 3 (c)). Here, it must be noted that also other external parameters influence the NO to $NO_2$ ratio, like e.g. atmospheric processing during plume transport as shown in Section 3.3 and Figure 7a. To account for this aging effect, we calculated initial NO to $NO_2$ ratios for each ship emission plume (i.e. the ratio that would have been observed close to the point of emission), using the respective plume age and the plume age dependence of the NO to $NO_2$ ratio as provided in Table 2. The resulting dependence of the NO to $NO_2$ ratio on the vessel speed, taken from the correlation plots (Figure 7a and Figure S13) is provided in Table 2 for both, the observed ratios ([NO]/[NO$_2$]$_{obs}$) and the calculated initial ratios ([NO]/[NO$_2$]$_{ini}$). The slopes of both regressions are identical, showing that additional influences on this ratio largely average out when binning the data or calculating the linear fits."

Fig3. Does organic EF vary with combustion efficiency or ship velocity?

*Reply: From our observations we do not see a significant dependence of the organic EF on ship velocity. While the BC EF decreases over the range of observed ship velocities by 70% of its initial value (observed at 0 m/s), no significant change of organic EF (+10% over the range of observed ship velocities with values ranging within an uncertainty interval of -13% to +33%) was observed as a function of ship velocity. However, the organics EF does depend strongly on fuel sulfur content as discussed in the manuscript (see Figure 4b).*

Fig.5 as eluded to in section 3.2 the importance of coagulation on aerosol number emission also depend on the plume age to some extent. that complexity is again not teased out here

*Reply: The reviewer is completely right: aerosol number emission factors do not only depend on coagulation within the exhaust system of the vessels (associated with ship size, i.e. gross tonnage), but also on coagulation within the transported plume and thus on dilution during emission into ambient air, associated with ambient wind speed (Figure 6b) and on transport time of the plume (Figure 9c). There are potentially more, albeit even smaller influences on PN emission factors like fuel quality or combustion efficiency, which are not presented here. Due to this enormous complexity of PN emission factors on many different influencing factors, we opted to present the individual dependencies of the strongest influences using correlation plots of binned data only. This averages out some of the other influences and allows focusing on the respective influence that is investigated. As an example, here we show the PN emission factor versus vessel gross tonnage (Figure 5), color-coded by plume age, as suggested by the reviewer:*

[Figure]

*This Figure shows that including this additional information within the plot does not provide additional clarity. Averaging into bins is needed to tease out the individual dependencies. This is why we used this approach in the manuscript. In order to make this additional dependence of PN emission factors on plume age, but also on ambient wind speed clearer, we added a reference to the other two figures to the discussion of Figure 5:*

*"… In agreement with this, we observe a reduction in particle number EF with increasing ship size (see Fig. 5), similar to the observations by Diesch et al. (2013). Additional influences on particle number EF are caused by coagulation in ambient air, either during plume emission from the stack (section 3.2, Figure 6b) or during plume transport (Section 3.3, Figure 9c). In the correlation shown in Figure 5, these influences are largely averaged out by using binned data. "*

Fig.6 besides coagulation and condensation, could wind speed or sea state alter the performances of the ships engines, and hence the aerosol emission factors?

*Reply: Thank you for this interesting suggestion. We do not see a significant dependence of the aerosol emission factors presented in Figure 6 on vessel speed and thus on ship engine performance. Furthermore, wind speed or sea state will probably have an influence on the load of the vessel's engines; however, this influence will depend on the relative direction of the wind and the vessel movement. Since these directions are independent from each other, they will likely average out for an ensemble of vessels.*

P 13 line 16. 1/3 of initial ratio? Initially it's almost all NO

*Reply: We appreciate this information. Since we did not perform measurements directly at the stack, we do not have any data on the NO/NO2 ratio at the time of emission. In our measurements this ratio was 0.6 for the youngest emission plume bin, as stated in the text. To make clear that we refer to the initial ratio as the one with the shortest atmospheric transport time we re-worded the respective sentence as follows:*

"We observe that the NO to $NO_2$ ratio decreases quickly down to 0.2 (i.e. one third of the ratio measured in the youngest plume age bin) during the first half hour of atmospheric transport of ship plumes emitted during daytime."

SO2/sO4 ratio decreases with humidity. Could this be partly due to cloud processing (if cloudy)?

*Reply: This is an interesting point. However, we can rule out cloud processing as the reason for the decrease in SO2/SO4 ratio: We did not observe relative humidity above 90% during the ship emission plume measurements and no cloud formation; furthermore, we observed this decrease in SO2/SO4 ratio over the whole range of observed relative humidity from 40% up to 90%.*

Fig9 changes in aerosol composition with aging. Would be interesting to see this separated to daytime and nighttime

*Reply: We agree with the reviewer's comment that separation of the aerosol composition changes with transport time into daytime and nighttime provides additional insight into the processes. When separating the data into daytime and nighttime data for the three different plume age intervals we observe that for some species in some plume age intervals we find only a very small number of emission factors above detection limit. To avoid an excessive influence of individual very large or small values we determined the typical composition during the individual plume age intervals by calculating median (instead of average) emission factors. This changes the individual fractional contributions slightly, but does not change the overall conclusion. In addition to the total (including daytime and nighttime data) aerosol composition dependence on plume age (in Figure 9d) we also present the data separated for daytime and nighttime in the supplementary material (Figure S14). All this and some additional information were included in the discussion of Figure 9 (Sect. 3.3):*
"To investigate the changes in chemical composition of the plume aerosol particles we calculated typical particle compositions for three different plume age intervals including approximately the same number of data points, using the medians of the respective emission factors: for plumes younger than 16 min, for plume ages between 16 and 40 min, and for plumes older than 40 min (see Fig. 9 (d)). The processes during plume aging are reflected in these relative compositions: The organic fraction contributes increasingly less and the inorganic fraction increasingly more to the particle phase. This is due to strong increases in the inorganic emission factors during plume transport, while the organic EF increases only very slightly. When looking separately at the plume development during day and night time, we find these effects to be much stronger during the day, compared to the night (see Figure S14), consistent with photochemical processes contributing to the formation of secondary inorganic aerosol components. We emphasise that the relatively low number of data points for some species during individual plume age intervals and large scatter of emission factors due to other influences results in some of the variability observed in the relative composition of the particle phase."

Relationship between O/C ratio and organic EF. Unclear whether 'increase in mass through Oxidation' occurs in atmosphere or in stack

*Reply: Thank you for pointing this out. This statement is related to processes in the transported plume. To make this clear we revised the sentence to (Sect. 3.2):*
"However, the particulate organic EF and the O/C ratio show a positive correlation (see Fig. 8 (b)), generally suggesting an increase of organic particulate mass, potentially through oxidation of gas phase organic material during plume transport."

P 18 line 19. Higher NO:NO2?

*Reply: Thank you for this important hint. We corrected the respective sentence.*